# Order within Chaos: Capturing Intrinsic Energy Anomalies for AI-Manipulated Image Forgery Localization

Yiming Wang[1]  Baiqi Wu[1]  Qingming Li[*1]  Jiahao Chen[1]  Tong Zhang[1]  Shouling Ji[1]

## Abstract

Recent advancements in generative AI have led to image editing models capable of producing realistic forgeries that evade traditional image forgery localization methods, as these approaches depend on physical noise absent in synthetic data. To address this challenge, we theoretically demonstrate that the diffusion process inherently suppresses local high-frequency variance, creating a statistical energy gap that is distinguishable from the natural entropy of optical imaging. Guided by this insight, we propose FLAME, a unified framework that utilizes a LAD map to capture these intrinsic anomalies, coupled with a parameter-efficient adapter for SAM to achieve precise, pixel-level forgery localization. Furthermore, to bridge the lag between forensic benchmarks and evolving generative models, we introduce EditStream, an automated pipeline for continuous, instruction-based training data synthesis. Extensive experiments demonstrate that FLAME establishes a new state-of-the-art, significantly outperforming previous methods on AI-generated forgery datasets while effectively generalizing to unseen generative architectures. Our code is available at https://github.com/phoenixnir/FLAME.

## 1. Introduction

Recent advancements in diffusion models have rendered synthetic images highly photorealistic and scalable (Rombach et al., 2022; Kawar et al., 2023). Beyond text-to-image systems that generate high-fidelity content, instruction-based editing further enables the manipulation of real images, thereby streamlining creative workflows and facilitating artistic expression (Huang et al., 2025b). However, the widespread availability of such powerful capabilities si-

multaneously amplifies the risks of visual deception, ranging from the proliferation of misinformation to identity fraud (Times, 2024; News, 2024). This emerging threat necessitates the evolution of **AI-Manipulated Image Forgery Localization (AI-IFL)**, which advances beyond image-level classification to pixel-level localization that explicitly delineates tampered regions for robust forensic verification (Kadha et al., 2025; Zou et al., 2025).

The primary objective of IFL is to localize manipulated regions within visual content. In the pre-AIGC era, IFL methodologies relied on verifying the consistency of physical signals. These approaches operated on the premise that conventional manipulations (Zanardelli et al., 2023) such as splicing, disrupt the continuity of the physical imaging pipeline, leaving discernible traces like sensor noise residuals and compression artifacts. However, the emergence of advanced diffusion models (e.g., Stable Diffusion (Podell et al., 2023), FLUX (Batifol et al., 2025), and DALL·E 3 (Betker et al., 2023)) has precipitated a fundamental paradigm shift in the forensic landscape. The dominant forensic cues have transitioned from physical anomalies to *synthetic traces* introduced by the generative process (Wang et al., 2023; Cao et al., 2025). Consequently, traditional methods often fail in this new context (Huang et al., 2025c).

To address the lack of physical signals, recent research (Xu et al., 2025; Huang et al., 2025c) has pivoted towards semantic-level scrutiny by leveraging MLLMs. These approaches treat forgery localization as a visual reasoning task, identifying inconsistencies such as geometric contradictions. However, this direction faces a semantic illusion trap. Up-to-date diffusion models possess such strong physical priors that they generate semantically impeccable edits, leaving little semantic contradiction to exploit. Furthermore, the tokenization process inherent to MLLMs limits their visual perception to a coarse granularity (Naseer et al., 2021; Song et al., 2024), rendering them blind to the subtle pixel-level imperfections that distinguish neural rendering from reality.

We argue that for the AI-IFL task, the most reliable forensic evidence lies neither in missing physical sensors nor in semantic errors, but in the **intrinsic statistical anomalies** induced by the generative mechanism itself. From a **statistical mechanics perspective**, diffusion models opti-

[1]Zhejiang University, Hangzhou, China. Correspondence to: Qingming Li <liqm@zju.edu.cn>.

*Proceedings of the 43rd International Conference on Machine Learning*, Seoul, South Korea. PMLR 306, 2026. Copyright 2026 by the author(s).

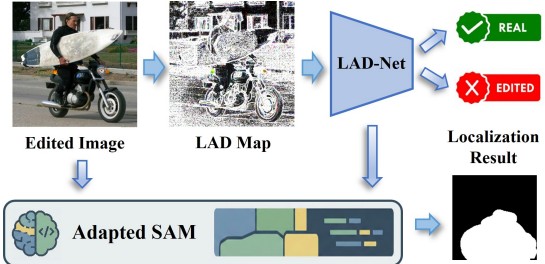

Figure 1. The overall workflow of FLAME.

mize an energy-based variational bound (Ho et al., 2020), which imposes a spectral bias: the denoising process tends to suppress high-frequency local variance to minimize energy (Rahaman et al., 2019). This creates a fundamental distinguishability: while authentic regions retain the natural chaos of optical imaging with high entropy, generated regions collapse into a state of artificial order with low entropy. Motivated by this observation, we formulate AI-IFL through Gibbs energy modeling (Derin & Elliott, 1987), which provides a principled way to characterize **pixel-level local dependency** structures. Here, pixel-level local dependency measures how each pixel statistically correlates with its neighborhood. Such local interactions can be expressed as spatial energy variations, allowing us to convert subtle distributional differences into spatially coherent indicators that facilitate precise localization.

Based on this local dependency viewpoint, our theoretical analysis identifies two diffusion-induced forensic signatures. First, generated regions often exhibit artificially regular local statistics, forming low-energy basins compared to authentic content. Second, latent-space editing may disrupt local correlations near tampering boundaries, leading to a localized energy surge that highlights forgery. To quantify these signatures, we introduce the Local Adjacency Discrepancy (LAD) map, which visualizes such statistical divergences and serves as a robust, content-agnostic forensic fingerprint.

Building upon the LAD map, we propose **Fine-grained Localization via Adjacency Map Energy (FLAME)**, a unified framework that implements a coarse-to-fine localization strategy. The architecture comprises two synergistic modules. First, a lightweight LAD-Net extracts artifact features from the LAD map to perform global image-level forgery detection and generate a preliminary coarse localization mask. Second, to achieve pixel-level precision, we incorporate a semantic refinement module based on SAM (Carion et al., 2025). By employing parameter-efficient adapters, this module aligns the low-level forensic cues with SAM's powerful segmentation priors, utilizing the coarse prediction as a spatial prompt to accurately delineate complex forgery boundaries. The overall workflow is shown in Figure 1.

Finally, to address the temporal obsolescence of static benchmarks, we introduce **EditStream**, a self-evolving data synthesis pipeline. By autonomously scouting and integrating emerging inpainting models, EditStream continuously updates the training distribution with novel artifact signatures. This mechanism fosters a virtuous cycle, compelling the model to learn architecture-invariant representations and ensuring robust generalization against unseen threats.

In summary, our contributions are as follows:

1. We theoretically prove that the inherent spectral bias of diffusion models inevitably induces distinct local energy anomalies, establishing a rigorous mathematical foundation for forgery localization.

2. We propose FLAME, a unified framework for AI-IFL that leverages these intrinsic artifacts. By utilizing the LAD map for feature extraction and a lightweight adapter-based SAM for refinement, we achieve precise forgery localization.

3. We develop EditStream, an automated pipeline for instruction-based image editing, capable of integrating the latest open-source models to generate diverse and challenging evaluation samples.

4. Leveraging EditStream, we construct a new dataset based on cutting-edge editing models. Comprehensive experiments validate that FLAME establishes a new state-of-the-art in accuracy and robustness across multiple benchmarks.

## 2. Related Work

**Image Forgery Localization.** IFL addresses the challenge of verifying image integrity by locating tampered regions. Traditional methods target conventional manipulations such as splicing and copy-move, relying on low-level physical forensic cues (Liu et al., 2022; Guillaro et al., 2023; Bazyleva et al., 2025). These approaches can be broadly divided into two categories. The first category focuses on detecting explicit artifacts derived from reconstruction errors (Bazyleva et al., 2025) or JPEG compression traces. The second targets implicit noise artifacts (Guillaro et al., 2023), where trained networks like Noiseprint (Cozzolino & Verdoliva, 2019) extract specific sensor residuals. While effective for physical manipulations, these methods suffer from significant performance degradation on AI-generated content (Huang et al., 2025c), as the generative process of diffusion models replaces camera sensor noise with synthetic signatures, rendering traditional physical cues obsolete.

To adapt to this paradigm shift, current research has harnessed large foundation models. Some work (Li et al., 2024; Zhu et al., 2025) utilizes the embedding space of vision encoders such as DINOv2 (Oquab et al., 2023) to detect subtle synthetic distribution shifts. Concurrently, a new branch employing MLLMs, such as SIDA (Huang et al.,

2025c) and FakeShield (Xu et al., 2025), exploits the text-to-image alignment nature of diffusion models to localize forgeries while providing linguistic explanations (Huang et al., 2025a). Despite these advancements, current methods often rely heavily on the generalizability of pre-trained encoders, potentially overlooking the intrinsic artifacts specific to the diffusion generation process (Cai et al., 2025).

**Localized Image Generation.** To benchmark IFL performance, the community has established several standard datasets (Dong et al., 2013). Early contributions relied on manually curated masks and prompts, limiting both scale and complexity (Zhang et al., 2023; Jia et al., 2023). Recent advancements introduce automated pipelines driven by MLLMs, which successfully enable large-scale data synthesis (Sun et al., 2024; Xu et al., 2025; Huang et al., 2025c). Despite this progress, existing generation protocols remain constrained by two limitations: a temporal lag behind the rapidly iterating editing models and a scarcity of editing diversity. These deficiencies compromise the alignment between benchmarks and real-world threats.

## 3. Theoretical Analysis

### 3.1. Preliminaries

Let $\mathbf{x} \in \mathbb{R}^{H \times W \times 3}$ be an RGB image. The Latent Diffusion Model (LDM) consists of a pre-trained variational autoencoder (Kingma et al., 2013) with an encoder $\mathcal{E}$, decoder $\mathcal{D}$, and time-conditional denoising network $\epsilon_\theta$ (Ho et al., 2020).

The image is first encoded into a reference latent representation $\mathbf{z}_0^{\text{ref}} = \mathcal{E}(\mathbf{x}) \in \mathbb{R}^{h \times w \times c}$. The generative process is modeled as a reverse Markov chain that progressively removes noise from a Gaussian prior $\mathbf{z}_T \sim \mathcal{N}(\mathbf{0}, \mathbf{I})$ to recover the latent signal. The transition kernel $p_\theta(\mathbf{z}_{t-1}|\mathbf{z}_t)$ is approximated by the denoiser $\epsilon_\theta(\mathbf{z}_t, t)$, which minimizes the reweighted evidence lower bound:

$$\mathcal{L}_{\text{LDM}} = \mathbb{E}_{\mathbf{z}_0, \epsilon, t} \left[ \|\epsilon - \epsilon_\theta(\mathbf{z}_t, t)\|_2^2 \right], \quad (1)$$

where $\mathbf{z}_t = \sqrt{\bar{\alpha}_t}\mathbf{z}_0 + \sqrt{1 - \bar{\alpha}_t}\epsilon$ represents the forward diffusion state at timestep $t$ derived from sample $\mathbf{z}_0$.

Localized editing aims to regenerate a specific region defined by a binary mask $\mathbf{m}$ (where $\mathbf{m} = 1$ indicates the edited region) while preserving the authentic context. This is achieved via masked reverse sampling. At each reverse step $t \rightarrow t - 1$, the latent state is composited as follows:

$$\mathbf{z}_{t-1} = \underbrace{\mathbf{m} \odot \hat{\mathbf{z}}_{t-1}^{\text{gen}}}_{\text{Generated}} + \underbrace{(1 - \mathbf{m}) \odot \mathbf{z}_{t-1}^{\text{ref}}}_{\text{Authentic}}, \quad (2)$$

where $\hat{\mathbf{z}}_{t-1}^{\text{gen}}$ is predicted by the denoiser $\epsilon_\theta$ given the current state, and $\mathbf{z}_{t-1}^{\text{ref}} \sim q(\mathbf{z}_{t-1}|\mathbf{z}_0^{\text{ref}})$ is a sample from the forward process of the original authentic image. Finally, the edited image is reconstructed via the decoder: $\hat{\mathbf{x}} = \mathcal{D}(\hat{\mathbf{z}}_0)$, where $\hat{\mathbf{z}}_0$ denotes the final state obtained from the reverse process.

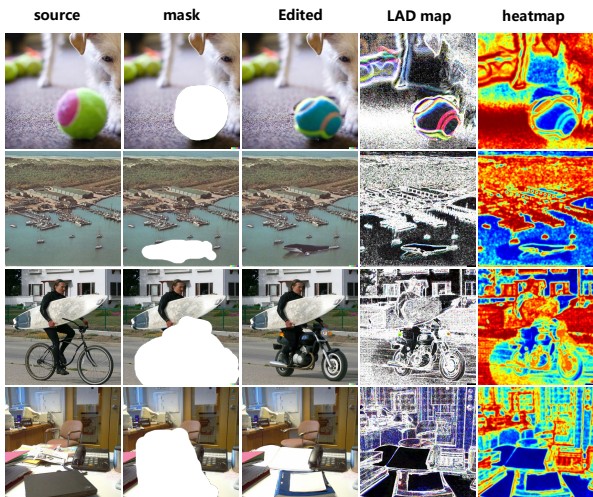

*Figure 2.* Visualization of the LAD map. From left to right, we display the source, ground-truth mask, and edited images, followed by the LAD map and its energy heatmap.

### 3.2. Artifact Separability via Gibbs Energy Modeling

To quantify the statistical divergence between authentic and synthetic patterns, we model the image lattice as a Gibbs Random Field (GRF) (Derin & Elliott, 1987). The probability distribution of an image $x$ is governed by $p(x) = \frac{1}{Z} \exp(-E(x))$, where $Z$ is the partition function and $E(x)$ represents the total energy state. We focus on the *local pairwise potential* $V(p, q)$ between adjacent pixels $p$ and $q$, defined as a function of their intensity magnitude difference:

$$V(p, q) = \rho \left( \|I(p) - I(q)\|_2 \right). \quad (3)$$

where $I(p)$ and $I(q)$ denote the RGB intensity vectors at pixels $p$ and $q$. This pairwise potential serves as a spatial domain proxy for spectral energy. A high potential value implies a large local gradient, which corresponds physically to the presence of high-frequency signal components. Conversely, a low potential indicates local smoothness, dominated by low-frequency structures. Based on this formulation, we establish the following assumption regarding the separability of authentic and generated distributions.

**Assumption 3.1** (Spectral-Energy Inequality)**.** Let $\mathcal{U}$ denote the authentic region and $\mathcal{M}$ denote the diffusion-generated region. We postulate that the expected local Gibbs energy in authentic regions strictly exceeds that of generated regions due to the disparity in high-frequency spectral density:

$$\mathbb{E}_{(p,q) \in \mathcal{U}}[V(p, q)] \geq \xi_{\text{noise}} > \mathbb{E}_{(p,q) \in \mathcal{M}}[V(p, q)]. \quad (4)$$

Here, $\xi_{\text{noise}}$ represents the irreducible energy floor induced by physical sensor noise.

*Remark* 3.2. The theoretical grounding for Assumption 3.1 stems from the fundamental difference between optical imaging physics and neural generative mechanisms:

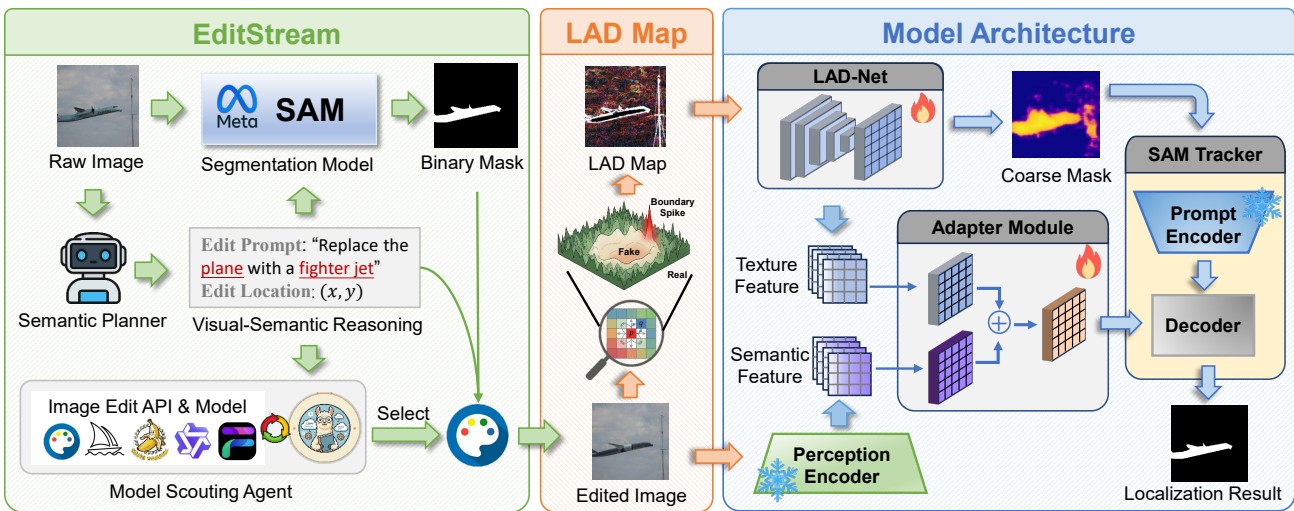

*Figure 3.* The overall flow of FLAME. **(Left)** EditStream functions as a self-evolving data engine, employing a Model Scouting Agent and Semantic Planner to generate diverse samples. **(Middle)** The LAD-Map Extraction module transforms the input image into an LAD map by capturing local dependency and energy potentials. **(Right)** The Model Architecture processes these inputs via LAD-Net to provide coarse guidance, which is then refined by a parameter-efficient Adapter Module within SAM to achieve pixel-level forgery localization.

**1. Physical Entropy vs. Neural Order:** Authentic images ($\mathcal{U}$) are captured via optical sensors where photon shot noise and thermal noise introduce stochastic, pixel-independent perturbations (Foi et al., 2008). These perturbations manifest as high-frequency irregularities, resulting in a high-entropy state with substantial local energy potentials.

**2. The Diffusion Spectral Bias:** In contrast, diffusion models generally optimize a reconstruction objective (e.g., MSE) constrained by a neural network architecture. As noted in spectral bias literature (Rahaman et al., 2019; Wang & Pehlevan, 2026), deep networks prioritize learning low-frequency semantic structures while struggling to capture high-frequency noise. Furthermore, the Lipschitz continuity of the VAE decoder acts as a low-pass filter, truncating the high-frequency tail of the spectrum. Consequently, generated regions ($\mathcal{M}$) collapse into a state of artificial order, statistically over-smoothed textures that lack the requisite high-frequency energy to match natural sensor noise.

**Theorem 3.3** (Energy Gap & Boundary Spike). *Let $\mathcal{E}_{loc}(p) = \mathbb{E}_{q \in \mathcal{N}_p}[\|I(p) - I(q)\|^2]$ be the expected local energy. Under Assumption 3.1 and latent mixing mechanism (Eq. 2):*

***1. Interior Gap:*** *The energy in the authentic region is greater than in the generated interior: $\mathcal{E}_{loc}(\mathcal{U}) > \mathcal{E}_{loc}(\mathcal{M})$.*

***2. Boundary Spike:*** *At the mask boundary $\partial\mathcal{M}$, the hard latent composite disrupts the local covariance structure, creating a manifold mismatch. This results in a localized surge in energy: $\mathcal{E}_{loc}(\partial\mathcal{M}) \gg \mathcal{E}_{loc}(\mathcal{M})$.*

Please refer to Appendix G.8 for full proof.

### 3.3. Local Adjacency Discrepancy Map

Building upon the theoretical *Energy Gap* and *Boundary Spike* in Theorem 3.3, we propose the LAD map. This representation is designed to quantify local Gibbs energy, translating theoretical spectral anomalies into a computable spatial fingerprint. Formally, the LAD map $\mathcal{L} \in \mathbb{R}^{H \times W}$ is computed by aggregating pairwise potentials within a local neighborhood. Specifically, for a pixel $p$, its value is calculated by the LAD operator:

$$\mathcal{L}(p) = \frac{1}{|\mathcal{N}_p|} \sum_{q \in \mathcal{N}_p} \tanh\left(\frac{\|I(p) - I(q)\|_2^2}{\tau^2}\right). \quad (5)$$

Here, $\mathcal{N}_p$ denotes the set of pixels within a hollow $n \times n$ spatial window centered at $p$ (where $n = 2m + 1$). We adopt $m = 1$ (i.e., $n = 3$) for efficient local estimation. The hyperbolic tangent *tanh* is employed to robustly saturate high-magnitude semantic gradients, strictly isolating the subtle noise residuals, with the sensitivity threshold controlled by $\tau$. Appendix G.9 provides a detailed discussion of the underlying mechanism of the LAD operator. By aggregating these robust pairwise potentials, the LAD map transforms the input image into a *Naturalness Response Field*. Authentic regions appear as high-energy areas due to preserved sensor noise, whereas manipulated regions manifest as low-energy voids, bounded by high-intensity spikes at the tampering trace, which aligns with our theoretical derivations. Figure 2 shows the visualization of the LAD map, further validating our statements.

# 4. Model Architecture

Building upon the extracted LAD map, we propose a coarse-to-fine forensic framework designed to locate manipulated regions with high precision. As illustrated in Figure 3, our architecture is composed of two synergistic modules: a lightweight **LAD Analysis Network (LAD-Net)** for initial texture-based detection, and a **SAM-based Semantic Refinement** module that leverages pre-trained semantic priors to delineate precise forgery boundaries.

## 4.1. Localization Architecture

**LAD Analysis Network.** The LAD map $\mathcal{L}$ highlights energy anomalies in the image. To further extract these cues, we employ a lightweight CNN as the backbone, which we call LAD-Net and denote as $f_{tex}$. This network serves three distinct purposes: extracting a texture-rich feature representation, predicting a global authenticity score, and generating a preliminary coarse localization mask. Let $F_{tex} = f_{tex}(\mathcal{L})$ be the multi-scale texture features extracted from the LAD map. These features are forwarded to two parallel branches:

**1. Global Classification Head:** This branch applies global average pooling followed by a fully connected layer to predict a scalar score $y_{cls} \in [0, 1]$, indicating the probability that the image has been manipulated.

**2. Coarse Localization Head:** A lightweight decoder projects $F_{tex}$ back to the spatial resolution, outputting a coarse mask $M_{coarse} \in [0, 1]^{H \times W}$. This mask roughly localizes the tampered area through diffusion artifacts, serving as a spatial prior for the subsequent refinement stage.

**SAM-based Semantic Refinement.** Although LAD-Net can effectively identify statistical anomalies, it cannot be directly relied upon for precise localization. This is because authentic images inherently contain low-energy regions (e.g., uniformly colored backgrounds) that are not indicative of manipulation. Therefore, a semantic-level analysis of the image is required to distinguish and exclude genuine low-energy regions. To address this, we integrate the *Segment Anything Model (SAM)* architecture (Kirillov et al., 2023). We adapt SAM (Carion et al., 2025) to the forgery localization task by conditioning it on both the texture anomalies and the coarse guidance from LAD-Net.

*Feature Fusion and Adaptation.* The target RGB image $I$ is processed by SAM's frozen image encoder (Bolya et al., 2025) to yield high-level semantic features, $F_{sem}$. Since $F_{sem}$ is optimized for natural object segmentation rather than forensics, we introduce a lightweight **Feature Adapter** to align it with the forensic domain. We fuse the semantic features with the texture features $F_{tex}$ from LAD-Net:

$$F_{adapted} = \text{Adapter}(F_{sem} \oplus F_{tex}), \qquad (6)$$

where $\oplus$ denotes channel-wise concatenation followed by a $1 \times 1$ convolution. The adapter, consisting of residual blocks, learns to modulate the semantic features with forensic cues, ensuring the model focuses on manipulated objects.

*Prompt-Guided Decoding.* Leveraging SAM's promptable design, we utilize the coarse mask $M_{coarse}$ as a dense spatial prompt. The mask is fed into SAM's prompt encoder to generate dense positional embeddings $E_{prompt}$. Finally, SAM's mask decoder takes the artifact-aware features $F_{adapted}$ and the prompt embeddings $E_{prompt}$ as input. It attends to the relevant semantic regions highlighted by LAD-Net and outputs the final, fine-grained forgery mask $M_{final}$.

## 4.2. Objective Function

We employ a hybrid objective function to jointly optimize pixel-level localization precision and image-level detection accuracy. The total loss $\mathcal{L}$ is formulated as:

$$L = L_{Dice} + \lambda_{focal} L_{focal}^{\alpha,\gamma} + \lambda_{IoU} L_{IoU} + \lambda_{det} L_{det}. \quad (7)$$

Here, $L_{Dice}$ and $L_{focal}$ drive the segmentation process, ensuring precise boundary delineation while mitigating the inherent class imbalance between small forged regions and the authentic background. Complementing these, $L_{IoU}$ supervises the mask quality prediction via $\ell_1$ regression, and $L_{det}$ enforces global semantic consistency through binary cross-entropy for image-level forgery classification. Following SAM conventions (Carion et al., 2025), we set $\lambda_{focal} = 20$ and $\lambda_{IoU} = 1$. We set $\lambda_{det} = 1$.

## 4.3. EditStream: Evolutionary Data Synthesis

A critical bottleneck in current AI forensics is the temporal lag where static training datasets fail to represent the artifacts of evolving generative models. To bridge this gap, we introduce **EditStream**, a self-evolving data synthesis framework that dynamically adapts to the latest text-to-image advancements. Unlike traditional static benchmarks, EditStream operates as a closed-loop adversarial generator to compel the FLAME localizer to learn architecture-invariant representations rather than overfitting to specific models. The framework is driven by a collaborative multi-agent system that automates the entire lifecycle of training data generation (see Appendix C for implementation details):

**1. Semantic Planning & Masking:** To ensure high-quality, logic-consistent edits, we employ a visual-semantic planner based on Qwen-VL (Yang et al., 2025) that analyzes scene semantics to generate editing instructions. These instructions are translated into precise pixel-level masks via SAM, guaranteeing that synthetic manipulations adhere to object boundaries without introducing background artifacts.

**2. Autonomous Model Scouting:** The core innovation of EditStream lies in its *Autonomous Model Scouting Agent*. Powered by Llama 3 (Grattafiori et al., 2024), this agent

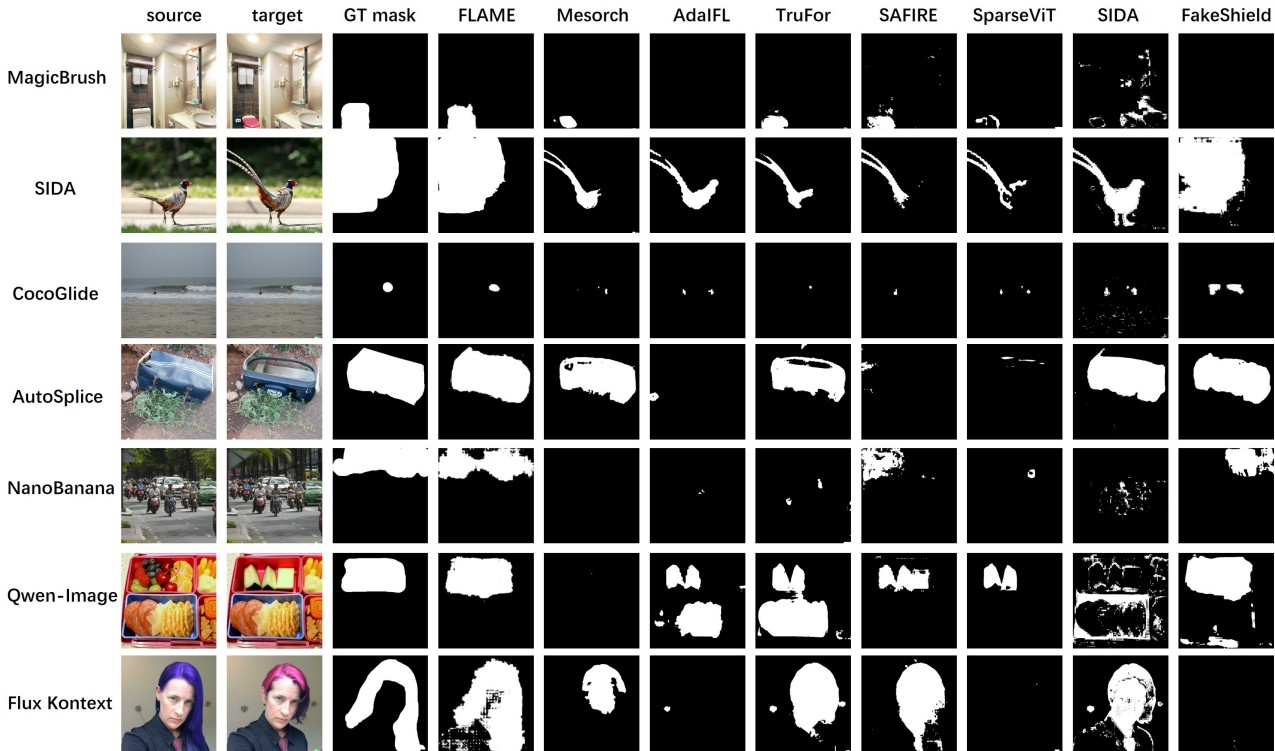

*Figure 4.* Overview of qualitative results across all models and datasets. Each row corresponds to a dataset sample and each column to the original and tampered images, the ground-truth mask, and each model's predicted mask.

actively monitors open-source repositories (e.g., Hugging Face) to identify trending inpainting models. It automatically parses model cards, synthesizes execution code, and integrates new checkpoints into the generation pipeline without human intervention.

By continuously injecting samples from the latest generative architectures, EditStream constructs a moving target for the detector. This evolutionary mechanism fosters a virtuous cycle: as generative models improve in realism, FLAME is continuously exposed to harder, more diverse artifact distributions, thereby sustaining its robustness in the wild.

# 5. Experiments

## 5.1. Experimental Settings

**Training Protocols.** We adopt a two-stage training strategy. The **Base FLAME** model is trained exclusively on established datasets, **MagicBrush** (Zhang et al., 2023) and a subset of **SID** (Huang et al., 2025c). To address emerging generative threats, the **Fine-tuned FLAME** adapts the base model with 500 images generated by the *EditStream* pipeline, specifically **Qwen-Image-Edit** (Wu et al., 2025). Crucially, to mitigate catastrophic forgetting during this adaptation, we implement a *direct replay strategy* (Zhou et al., 2024): each training batch mixes 20% of the origi-

nal training data (MagicBrush and SID) with the new fine-tuning samples.

**Testing Protocols.** Our evaluation enforces a separation between distributions. For the Base FLAME, **In-Distribution (ID)** performance is assessed on the test splits of MagicBrush and SID. Conversely, **CoCoGLIDE** (Guillaro et al., 2023), **AutoSplice** (Jia et al., 2023), and all three EditStream-generated datasets (**Nano Banana** (Comanici et al., 2025), **Qwen-Image-Edit**, and **Flux.1 Kontext** (Batifol et al., 2025)) serve as **Out-Of-Distribution (OOD)** benchmarks. For the Fine-tuned FLAME, the **Qwen-Image-Edit** dataset is reclassified as ID following its inclusion in the adaptation phase, while the remaining datasets retain their OOD status to evaluate cross-domain generalization.

**Baselines and Metrics.** We benchmark against a suite of state-of-the-art methods: **TruFor** (Guillaro et al., 2023), **SAFIRE** (Kwon et al., 2025), **Mesorch** (Zhu et al., 2025), **AdaIFL** (Li et al., 2024), **SparseViT** (Su et al., 2025), **SIDA** (Huang et al., 2025c), and **FakeShield** (Xu et al., 2025). Performance is measured using pixel-level **Mean IoU** and **F1 Score** for localization, and image-level **Accuracy** (ACC) and **Average Precision** (AP) for detection capabilities. Unless otherwise specified, aggregate comparisons are based on OOD scores to assess cross-domain generalization. See Appendix D for details. Since AI-IFL

*Table 1.* Quantitative comparison of pixel-level localization. FLAME refers to the base model, while FLAME-F denotes the version fine-tuned via our EditStream pipeline. Underlined values indicate ID evaluation. The Average column calculates the mean performance across the last five datasets (from CoCoGLIDE to Flux Kontext) to assess generalization capabilities against OOD datasets.

| Model | MagicBrush | | SID | | CoCoGLIDE | | AutoSplice | | NanoBanana | | Qwen-Image | | Flux Kontext | | Average | |
|---|---|---|---|---|---|---|---|---|---|---|---|---|---|---|---|---|
| | IoU↑ | F1↑ | IoU↑ | F1↑ | IoU↑ | F1↑ | IoU↑ | F1↑ | IoU↑ | F1↑ | IoU↑ | F1↑ | IoU↑ | F1↑ | IoU↑ | F1↑ |
| SAFIRE | 0.297 | 0.485 | 0.214 | 0.274 | 0.394 | 0.467 | 0.192 | 0.251 | 0.114 | 0.153 | 0.217 | 0.269 | 0.190 | 0.225 | 0.221 | 0.273 |
| Mesorch | 0.150 | 0.211 | 0.124 | 0.219 | 0.382 | 0.450 | 0.210 | 0.283 | 0.102 | 0.139 | 0.216 | 0.286 | 0.124 | 0.180 | 0.207 | 0.268 |
| TruFor | 0.281 | 0.391 | 0.188 | 0.243 | 0.371 | 0.457 | 0.364 | 0.483 | 0.071 | 0.092 | 0.228 | 0.312 | 0.203 | 0.276 | 0.247 | 0.324 |
| AdaIFL | 0.122 | 0.215 | 0.128 | 0.190 | 0.209 | 0.266 | 0.227 | 0.337 | 0.091 | 0.126 | 0.066 | 0.092 | 0.073 | 0.104 | 0.133 | 0.185 |
| SIDA | 0.106 | 0.180 | 0.488 | 0.565 | 0.375 | 0.465 | 0.393 | 0.483 | 0.005 | 0.012 | 0.089 | 0.143 | 0.092 | 0.149 | 0.191 | 0.250 |
| FakeShield | 0.091 | 0.126 | 0.117 | 0.137 | 0.138 | 0.150 | 0.238 | 0.296 | 0.086 | 0.095 | 0.098 | 0.110 | 0.096 | 0.108 | 0.131 | 0.152 |
| SparseViT | 0.087 | 0.154 | 0.185 | 0.227 | 0.325 | 0.386 | 0.201 | 0.279 | 0.021 | 0.049 | 0.056 | 0.073 | 0.048 | 0.061 | 0.130 | 0.170 |
| **FLAME (ours)** | 0.538 | 0.650 | 0.580 | 0.677 | 0.469 | 0.576 | 0.501 | 0.624 | 0.216 | 0.295 | 0.321 | 0.408 | 0.285 | 0.391 | 0.358 | 0.459 |
| **FLAME-F†(ours)** | 0.507 | 0.632 | 0.569 | 0.650 | 0.481↑ | 0.602↑ | 0.498 | 0.618 | 0.391↑ | 0.454↑ | 0.482↑ | 0.603↑ | 0.446↑ | 0.548↑ | 0.460↑ | 0.565↑ |

*Table 2.* Quantitative comparison of image-level forgery detection. The experimental setup is consistent with Table 1.

| Model | MagicBrush | | SID | | CoCoGLIDE | | AutoSplice | | NanoBanana | | Qwen-Image | | Flux Kontext | | Average | |
|---|---|---|---|---|---|---|---|---|---|---|---|---|---|---|---|---|
| | ACC↑ | AP↑ | ACC↑ | AP↑ | ACC↑ | AP↑ | ACC↑ | AP↑ | ACC↑ | AP↑ | ACC↑ | AP↑ | ACC↑ | AP↑ | ACC↑ | AP↑ |
| SAFIRE | 0.525 | 0.640 | 0.596 | 0.570 | 0.481 | 0.481 | 0.462 | 0.623 | 0.570 | 0.592 | 0.454 | 0.644 | 0.616 | 0.492 | 0.517 | 0.566 |
| Mesorch | 0.630 | 0.693 | 0.625 | 0.648 | 0.606 | 0.713 | 0.558 | 0.597 | 0.468 | 0.492 | 0.598 | 0.749 | 0.547 | 0.647 | 0.555 | 0.640 |
| TruFor | 0.675 | 0.776 | 0.531 | 0.558 | 0.604 | 0.650 | 0.626 | 0.693 | 0.494 | 0.498 | 0.600 | 0.679 | 0.578 | 0.656 | 0.580 | 0.635 |
| AdaIFL | 0.526 | 0.593 | 0.546 | 0.572 | 0.519 | 0.547 | 0.546 | 0.575 | 0.474 | 0.483 | 0.547 | 0.562 | 0.523 | 0.549 | 0.522 | 0.543 |
| SIDA | 0.812 | 0.824 | 0.725 | 0.768 | 0.606 | 0.710 | 0.541 | 0.743 | 0.437 | 0.350 | 0.526 | 0.543 | 0.522 | 0.537 | 0.526 | 0.577 |
| FakeShield | 0.664 | 0.671 | 0.632 | 0.648 | 0.532 | 0.560 | 0.574 | 0.607 | 0.470 | 0.491 | 0.582 | 0.617 | 0.493 | 0.528 | 0.530 | 0.561 |
| SparseViT | 0.486 | 0.487 | 0.511 | 0.555 | 0.508 | 0.508 | 0.572 | 0.478 | 0.508 | 0.523 | 0.541 | 0.607 | 0.490 | 0.553 | 0.529 | 0.534 |
| **FLAME (ours)** | 0.901 | 0.917 | 0.916 | 0.922 | 0.732 | 0.789 | 0.714 | 0.763 | 0.715 | 0.747 | 0.781 | 0.803 | 0.754 | 0.781 | 0.639 | 0.677 |
| **FLAME-F†(ours)** | 0.893 | 0.905 | 0.911 | 0.925 | 0.754↑ | 0.801↑ | 0.698 | 0.756 | 0.812↑ | 0.805↑ | 0.821↑ | 0.845↑ | 0.792↑ | 0.828↑ | 0.715↑ | 0.755↑ |

requires localizing the edited subset within an otherwise authentic image, we evaluate methods that produce pixel-level masks. Image-level synthetic-image detectors are therefore outside the primary localization comparison, while Table 2 reports detection scores derived from localization outputs.

## 5.2. Evaluation Results of Base FLAME

**Localization Results.** We first evaluate the pixel-level localization performance of the Base FLAME model. As presented in Table 1, our method establishes a new state-of-the-art, delivering the highest average performance across all OOD benchmarks in terms of both F1 and IoU scores. This superiority is particularly pronounced in the zero-shot generalization setting against emerging generative models (Nano Banana, Qwen-Image, and Flux). Existing methodologies typically rely on traditional IFL cues or target the conspicuous artifacts characteristic of early-stage generators, leading to significant performance degradation under this domain shift. In contrast, FLAME maintains robust localization capabilities. This empirical evidence validates our theoretical premise: the statistical energy anomalies captured by the LAD map are intrinsic to the diffusion process itself, rather than being specific to a particular model architecture or dataset distribution.

Qualitatively, Figure 4 visualizes the predicted masks. Compared to baselines, which often produce fragmented regions or fail to detect subtle manipulations, FLAME generates precise masks that align with the ground truth, accurately

delineating forgeries even in scenarios involving complex background interactions or varying object scales.

**Detection Results.** Beyond localization, we evaluate the image-level ability to distinguish between authentic and forged images. For baselines not explicitly designed for this task, we adopt the maximum value of the localization map as the detection statistic, as we empirically observed that it yields superior discrimination compared to mean pooling. Table 2 presents the Accuracy and AP scores. Consistent with the localization findings, FLAME achieves state-of-the-art detection performance. The high AP scores indicate that our model maintains a reliable ranking capability, effectively assigning higher anomaly scores to forged samples regardless of the specific generative architecture used.

## 5.3. Adaptability of Fine-tuned FLAME

We analyze the performance of Fine-tuned FLAME (FLAME-F) based on the results in Tables 1 and 2. The performance shifts reveal three critical capabilities:

**Rapid Target Adaptation.** FLAME-F exhibits a significant performance leap on the target Qwen-Image dataset, which validates that the synthetic samples from EditStream successfully capture the distinct artifact signatures of modern generative models, enabling the network to effectively align its feature space with the evolving distribution.

**Cross-Architecture Generalization.** Improvements extend beyond the target domain. Despite being fine-tuned exclu-

*Table 3.* Quantitative comparison results under different perturbations.

| Perturbation | MagicBrush | | SID | | CoCoGLIDE | | AutoSplice | | NanoBanana | | Qwen-Image-Edit | | Flux Kontext | | Average | |
|---|---|---|---|---|---|---|---|---|---|---|---|---|---|---|---|---|
| | IoU↑ | F1↑ | IoU↑ | F1↑ | IoU↑ | F1↑ | IoU↑ | F1↑ | IoU↑ | F1↑ | IoU↑ | F1↑ | IoU↑ | F1↑ | IoU↑ | F1↑ |
| JPEG | 0.267 | 0.380 | 0.425 | 0.507 | 0.413 | 0.529 | 0.486 | 0.601 | 0.208 | 0.288 | 0.216 | 0.312 | 0.209 | 0.302 | 0.329 | 0.417 |
| Gaussian Blur | 0.374 | 0.498 | 0.568 | 0.653 | 0.420 | 0.535 | 0.498 | 0.622 | 0.202 | 0.285 | 0.319 | 0.405 | 0.275 | 0.381 | 0.379 | 0.483 |
| Gaussian Noise | 0.216 | 0.305 | 0.374 | 0.473 | 0.330 | 0.438 | 0.420 | 0.563 | 0.197 | 0.282 | 0.179 | 0.272 | 0.175 | 0.263 | 0.270 | 0.371 |
| No Perturbation | **0.538** | **0.650** | **0.580** | **0.677** | **0.469** | **0.576** | **0.501** | **0.624** | **0.216** | **0.295** | **0.321** | **0.408** | **0.285** | **0.391** | **0.416** | **0.517** |

*Table 4.* Ablation study of different components.

| Configuration | IoU↑ | F1↑ |
|---|---|---|
| w/o LAD Map | 0.294 | 0.380 |
| w/o Adapter | 0.313 | 0.392 |
| w/o SAM | 0.379 | 0.458 |
| **FLAME** | **0.567** | **0.662** |

*Table 5.* Ablation study on different operators and kernel sizes.

| Operator | $3 \times 3$ | | $5 \times 5$ | | $7 \times 7$ | |
|---|---|---|---|---|---|---|
| | IoU↑ | F1↑ | IoU↑ | F1↑ | IoU↑ | F1↑ |
| Min | 0.439 | 0.525 | 0.477 | 0.568 | 0.289 | 0.355 |
| Max | 0.451 | 0.542 | 0.445 | 0.531 | 0.310 | 0.384 |
| Avg | 0.502 | 0.596 | 0.478 | 0.572 | **0.458** | **0.545** |
| **LAD** | **0.567** | **0.662** | **0.521** | **0.615** | 0.457 | 0.542 |

sively on Qwen-generated data, FLAME-F demonstrates enhanced performance on other unseen contemporary models, such as Flux.1 and Nano Banana. This indicates that the model avoids overfitting to specific generator noise. Instead, it leverages the fresh data to learn a generalized representation of artifacts common to advanced diffusion architectures.

**Mitigation of Forgetting.** FLAME-F maintains high stability on historical benchmarks with negligible performance degradation. This confirms that our framework successfully balances plasticity, adapting to new domains, with stability, ensuring that the integration of emerging threats does not compromise defense capabilities.

## 5.4. Robustness Evaluation

In real-world scenarios, images often undergo post-processing operations that attenuate subtle forensic traces. We therefore evaluate FLAME across all benchmarks under three common perturbations: JPEG compression (quality level 75), Gaussian blur (kernel size 3), and Gaussian noise (variance 5), with results summarized in Table 3. Performance decreases under all perturbations, which is expected because LAD is derived from local high-frequency statistics. The degradation is larger under JPEG compression and Gaussian noise: the former quantizes high-frequency coefficients, while the latter injects global variance, both of which obscure the energy contrast between authentic and generated regions. In contrast, Gaussian blur causes a smaller drop in our setting, suggesting that the region-level energy gap and boundary mismatch can remain informative even when the finest texture residuals are attenuated. Overall, these results show that the proposed statistical cue retains localization utility under common post-processing, while transformations that substantially rewrite local statistics remain more challenging.

## 5.5. Ablation Study

**Efficacy of Key Components.** We assess the contribution of core modules by comparing the full model against three variants: (1) **w/o SAM Adapter**, which removes the feature fusion module; (2) **w/o LAD Map**, which replaces the Local Adjacency Discrepancy map with the raw RGB input; and (3) **w/o SAM**, which bypasses the SAM refinement to use the coarse difference mask directly. All ablations are trained on the training sets of SID and MagicBrush and evaluated on their validation splits. As shown in Table 4, performance degrades in all ablated settings. The most significant drop occurs without the LAD map, confirming that explicit artifact modeling is crucial for detecting subtle diffusion traces. Additionally, removing the SAM-related components leads to coarse, blocky predictions, underscoring their importance for pixel-perfect boundary localization.

**Impact of Neighborhood Size and Operators.** We further investigate the influence of the local aggregation operator and the neighborhood window size $k \in \{3, 5, 7\}$. As detailed in Table 5, the proposed LAD operator consistently outperforms standard pooling mechanisms. While Min and Max suffer from sensitivity to outliers and Avg suppresses high-frequency forensic traces, LAD utilizes a robust potential function to capture the intrinsic energy anomalies.

Regarding window size, we observe a monotonic performance decline as $k$ increases, confirming that larger kernels over-smooth local pixel variances. Consequently, the $3 \times 3$ LAD configuration proves optimal for preserving the subtle spectral signatures required for precise localization.

For ablation studies on individual parameters, please refer to the Appendix E.

# 6. Conclusion

In this work, we propose FLAME, an AI-IFL framework grounded in the theoretical discovery of diffusion-induced spectral bias. Our approach utilizes a lightweight LAD-Net to capture intrinsic energy anomalies from the LAD map for preliminary localization. This coarse prediction serves as a spatial prompt for the SAM architecture, into which we integrate a compact adapter to fuse forensic cues for precise refinement. Furthermore, we introduce EditStream, a self-evolving data synthesis pipeline that addresses the limitations of static benchmarks by continuously integrating emerging generative models. Evaluations demonstrate that FLAME establishes a new state-of-the-art, effectively generalizing across unseen generative architectures.

# Acknowledgements

This work was partly supported by the New Generation Artificial Intelligence-National Science and Technology Major Project under No. 2025ZD0123503, NSFC under No. U2441239 and U24A20336 and No. 62502433, the China Postdoctoral Science Foundation under No. 2024M762829, 2025M781523 and 2025M781522, Zhejiang Key Laboratory of Decision Intelligence under No. 2025E10006, Zhejiang Provincial Natural Science Foundation Exploration of China under No. LMS26F020003, State Key Laboratory of Cryptography and Digital Economy Security under No. KFYB2504, the Zhejiang Provincial Natural Science Foundation under No. LD24F020002, and the "Pioneer and Leading Goose" R&D Program of Zhejiang under No. 2025C02033 and 2025C01082.

# Impact Statement

This paper aims to advance the field of AI forensics and digital media safety. As generative diffusion models become increasingly accessible and capable of producing photorealistic edits, the risks associated with visual deception have escalated significantly. Our work, FLAME, contributes a robust defense mechanism by theoretically identifying and localizing intrinsic statistical anomalies in AI-manipulated content, thereby restoring trust in digital media.

Furthermore, we introduce EditStream, an automated pipeline for synthesizing training data. While we acknowledge that automated generation tools carry a potential dual-use risk, our deployment of EditStream is strictly intended for adversarial robustness. It enables forensic models to evolve alongside rapid advancements in generative architectures. By establishing a virtuous cycle where defenses are continuously updated against emerging threats, our work seeks to mitigate the arms race in digital forgery and foster a safer information ecosystem.

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

# A. Limitations

FLAME is most effective when AI-edited regions leave detectable local statistical discrepancies and provide a usable boundary cue for refinement. Its performance can degrade in naturally smooth or repetitive authentic regions, where LAD responses may be difficult to distinguish from generated content and can produce diffuse coarse proposals or false positives. Very small edits, weak object boundaries, and scenes containing many similar candidate objects may also limit SAM-based refinement. Among editing operations, object removal is particularly challenging because inpainting often synthesizes background texture that is close to the surrounding authentic context and may not introduce a clear semantic transition. Finally, severe post-processing, extreme ISP denoising, or detector-aware anti-forensic perturbations can substantially alter the local statistics used by the LAD branch. Addressing these cases will require stronger context modeling and robustness-aware training in future AI forgery localization systems.

# B. Additional Visualizations

In this section, we provide a comprehensive qualitative analysis to empirically validate the efficacy of the FLAME framework. Figure 5 visualizes the intermediate representations generated at each stage of the pipeline, tracking the transformation from raw statistical artifacts to precise semantic segmentation.

Columns 4 and 5 corroborate our theoretical analysis presented in Theorem 3.3. As observed, authentic regions maintain high-frequency variations, appearing as high-energy noise, whereas diffusion-generated regions collapse into a state of artificial order and appear as low-energy voids.

Column 6 displays the *Coarse Mask* predicted by the LAD-Net. By processing the LAD map, the LAD-Net successfully identifies the general location of the forgery, effectively acting as a global attention mechanism. However, relying solely on statistical anomalies can yield blob-like predictions that lack boundary precision. Column 7 (*Results*) demonstrates the contribution of our SAM-based refinement module. Using the coarse mask as a spatial prompt, the adapter leverages SAM's powerful segmentation priors to snap the prediction to the object's semantic boundaries. This coarse-to-fine strategy ensures that FLAME achieves pixel-level accuracy even when the statistical traces at the boundary are faint or disrupted by compression.

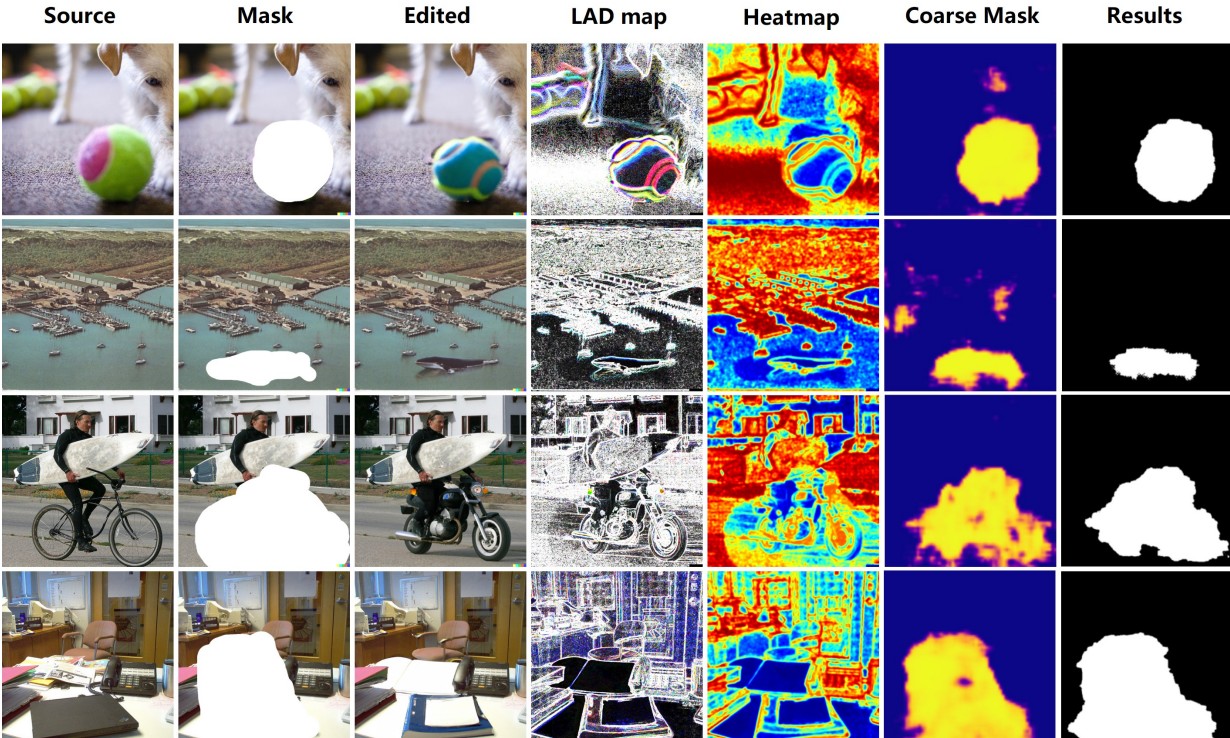

*Figure 5.* Step-by-step visualization of the FLAME pipeline.

# C. EditStream: Implementation and Data Generation Details

This section provides a comprehensive elucidation of the EditStream pipeline, complementing the high-level overview presented in Section 4.3. We detail its architectural mechanics and the rigorous protocols employed for dataset creation.

## C.1. Visual-Semantic Reasoning and Mask Generation

The first stage of the pipeline focuses on understanding the scene semantics to determine *where* and *what* to edit. We leverage the multimodal capabilities of **Qwen3-VL-8B-Instruct** (Yang et al., 2025) as the semantic planner. Given a raw input image, the MLLM is prompted to analyze the scene layout and identify salient objects or background regions suitable for manipulation. It outputs a structured description of the target region along with approximate spatial coordinates.

To translate these high-level semantic descriptions into precise pixel-level signals, we utilize SAM (Carion et al., 2025). By injecting the text descriptions and spatial coordinates provided by the MLLM into SAM's promptable interface, we obtain high-fidelity binary masks that strictly adhere to object boundaries, minimizing boundary artifacts during the subsequent inpainting phase.

## C.2. Autonomous Model Scouting Agent

To mitigate the temporal lag between the emergence of new generative models and their inclusion in forensic benchmarks, we engineer the Autonomous Model Scouting Agent. This agent is not merely a static script but a dynamic system powered by Llama 3 (Grattafiori et al., 2024), capable of reasoning, tool usage, and code synthesis.

**Tool-Augmented Search Mechanism.** Unlike passive crawlers, our agent interacts directly with the ecosystem via the official Hugging Face API, which we encapsulate as a structured tool, `search_hf_models()`.

1. **Query Formulation:** The agent leverages the Hugging Face API, selecting the image-to-image task category to retrieve the most recent and widely adopted open-source image editing models.

2. **Multi-Metric Filtering:** The raw search results are filtered through a composite scoring function. This function weighs quantitative metrics, download counts, likes, and trending status, alongside qualitative indicators such as "recently updated" timestamps. This ensures that the selected models are not only popular but represent the current state-of-the-art.

3. **Candidate Selection:** Based on this analysis, the agent autonomously selects the top-3 most promising candidates that are architecturally distinct from the existing pool.

**Dynamic Code Synthesis and Integration.** Once a candidate model is selected, the agent must integrate it into the EditStream pipeline without human engineering.

- **Model Card Parsing:** The agent retrieves and analyzes the unstructured text within the candidate's Model Card (README). It extracts critical metadata: input constraints (e.g., resolution limits), dependency requirements, and the specific input format.

- **Execution Logic Generation:** Leveraging its code-generation capabilities, Llama 3 synthesizes a bespoke Python wrapper for the model. This wrapper standardizes the input-output interface, ensuring that diverse models, regardless of their internal implementation, conform to the EditStream pipeline's expected protocols.

- **Sandboxed Verification:** The synthesized code undergoes a dry-run verification in a sandboxed environment to ensure execution stability before deployment.

For detailed prompts of different components, please refer to Appendix I.

## C.3. Dataset Creation Details

To construct a robust evaluation benchmark, we adhere to a strict data generation protocol that ensures diversity in semantic content, editing operations, and manipulation footprints.

**Complexity-Aware Source Selection.** We curate a pool of high-resolution authentic images sourced from open-domain datasets (e.g., subsets of LAION-5B (Schuhmann et al., 2022) and COCO (Lin et al., 2014)). To avoid trivial cases, we implement a multi-stage filtering protocol inspired by scene complexity heuristics:

1. **Aesthetic Filtering:** We retain images with an aesthetic score $> 6.0$ to ensure high visual fidelity.

2. **Semantic Density:** We require the presence of at least three distinct object classes (detected via a pre-trained detector), ensuring the scene offers rich semantic context for manipulation.

3. **Spatial Constraints:** To prevent single-object dominance, we exclude images where the largest bounding box exceeds 50% of the total canvas area. This ensures the resulting forgeries involve realistic multi-object interactions rather than simple foreground replacements.

**Balancing Editing Operations.** A core objective is to evaluate robustness across different manipulation types. We instruct the Semantic Planner (Qwen-VL) to categorize its editing instructions into four distinct operations:

1. **Object Addition:** Inserting new objects into the scene (e.g., "Add a red car in the driveway").

2. **Object Removal:** Erasing existing entities (e.g., "Remove the pedestrian").

3. **Object Replacement:** Swapping an object with a semantically different one (e.g., "Replace the dog with a cat").

4. **Background Alteration:** Modifying environmental textures or styles (e.g., "Change the grass to snow").

The planner is prompted to sample these operations with a uniform probability distribution, ensuring balanced coverage across all four categories. To address this, we implement a *dynamic priority mechanism*:

1. **Feasibility Analysis:** For each source image, the Semantic Planner first identifies the subset of feasible operations that respect the scene's semantic and spatial constraints.

2. **Frequency-Inverse Sampling:** From this feasible subset, the system selects the operation that is currently least represented in the generated dataset buffer. This active balancing strategy prevents the path of least resistance from dominating the distribution, ensuring approximately uniform coverage ($\sim$25%) across all manipulation types.

**Inpainting Area Distribution.** To prevent the localizer from overfitting to specific mask sizes, we enforce a stratified sampling strategy for the manipulated region's size. The generated masks vary from small-scale (occupying $< 5\%$ of image area) to large-scale ($> 30\%$), challenging the model's ability to detect both subtle and dominant forgeries.

**Admission-Control Filtering.** Before inserting generated samples into the dataset, EditStream applies a two-stage admission-control filter to reduce low-quality or semantically inconsistent edits. First, a deterministic pre-check verifies mask validity, area ratio, and boundary alignment, discarding degenerate masks or edits with negligible visible changes. Second, a VLM-based verifier evaluates instruction fidelity, semantic consistency, and edit quality. Samples that fail either stage are discarded or regenerated, complementing the planning-stage safeguards with explicit sample-level verification.

**Dataset Composition and Model-Specific Handling.** The final dataset consists of 6,500 images generated using three distinct architectures:

- **Qwen-Image-Edit (Wu et al., 2025):** We generated 3,000 samples. This model follows the standard inpainting paradigm, accepting a user-provided binary mask and text prompt. We fine-tune FLAME on 500 samples and test on 2,500 samples.

- **Flux.1 Kontext (Batifol et al., 2025):** We generated 3,000 samples. Similar to Qwen, this model utilizes the masks generated by our SAM-based Semantic Planner.

- **NanoBanana (Comanici et al., 2025):** We generated 500 samples. A unique challenge with NanoBanana is that it operates as an instruction-to-image editor and does not support external mask injection. Instead, it autonomously determines the region to edit based on the textual prompt. To accommodate this, we inverted the workflow: we provide the edit instruction, and the model returns both the manipulated image and the mask where the image changed. We utilize this returned mask as the ground truth for training and evaluation.

# D. Detailed Experimental Settings

In this section, we provide a granular breakdown of the experimental setup to ensure reproducibility. We detail the dataset compositions, the specific definitions of evaluation metrics, and the implementation hyperparameters used for the FLAME framework.

## D.1. Datasets

To comprehensively evaluate the robustness and generalizability of FLAME, we curate a diverse collection of benchmarks encompassing both manually annotated edits and large-scale synthetic manipulations. The datasets are categorized into ID sets used for training/validation, and OOD sets used exclusively for zero-shot testing.

- **MagicBrush** (Zhang et al., 2023): This dataset represents high-quality, instruction-guided image editing. It contains diffusion-based edits produced via DALL-E 2 (Ramesh et al., 2022), verified through rigorous human annotation. The dataset features multiple editing turns per image; we compute the ground-truth binary mask as the union of forged pixels across all rounds. The final composition includes 8,807 samples. We strictly follow the official split, utilizing the training set for model training, 528 samples for validation, and 1,053 samples for ID testing.

- **SID** (Huang et al., 2025c): A large-scale corpus comprising roughly 100,000 edits generated via Latent Diffusion Models (LDM) (Rombach et al., 2022). Given the sheer volume, we sample a subset to balance the training distribution. Specifically, we utilize 10,000 tampered samples for training and 528 for validation. For evaluation, we employ the full tampered test set to assess performance on LDM-based forgeries.

- **AutoSplice** (Jia et al., 2023): This dataset focuses on text-prompt manipulated images using DALL-E 2. We treat the entire dataset as an OOD benchmark, allocating all 3,621 images exclusively for testing.

- **CoCoGLIDE** (Guillaro et al., 2023): A challenging evaluation set containing 512 images edited using the GLIDE model. We utilize all 512 samples for OOD testing to evaluate resilience against GLIDE-specific artifacts.

For datasets generated by EditStream, please refer to Appendix C.3 for details.

## D.2. Evaluation Metrics

We employ standard forensic metrics to quantify performance at both the pixel-level localization and image-level detection.

### D.2.1. PIXEL-LEVEL LOCALIZATION

Localization performance is measured by comparing the predicted binary mask $M_{pred}$ (obtained by thresholding the probability map at 0.5) with the ground-truth mask $M_{gt}$.

- **Intersection over Union (IoU):** Measures the overlap between the predicted and ground-truth forged regions.

$$IoU = \frac{|M_{pred} \cap M_{gt}|}{|M_{pred} \cup M_{gt}|} = \frac{TP}{TP + FP + FN} \tag{8}$$

  where $TP$, $FP$, and $FN$ denote True Positives, False Positives, and False Negatives, respectively.

- **F1 Score:** The harmonic mean of precision and recall, providing a balanced view of localization accuracy, especially for small forged regions.

$$F1 = \frac{2 \cdot Precision \cdot Recall}{Precision + Recall} = \frac{2TP}{2TP + FP + FN} \tag{9}$$

### D.2.2. IMAGE-LEVEL DETECTION

For the binary classification task (Tampered vs. Authentic), we derive an image-level anomaly score $y_{score}$ from the predicted mask. We empirically found that the maximum activation value yields the best separation: $y_{score} = \max_{(i,j)} M_{prob}(i, j)$.

- **Accuracy (ACC):** The ratio of correctly classified images to the total number of images, using a decision threshold of 0.5.

- **Average Precision (AP):** The area under the Precision-Recall curve, providing a threshold-independent measure of the detector's discriminative ability.

## E. Additional Experimental Results

In this section, we provide a granular analysis of the hyperparameters, robustness, and generalization capabilities of FLAME, supplementing the main experimental results.

### E.1. Sensitivity to Threshold $\tau$

The temperature parameter $\tau$ in the LAD operator (Eq. 5) plays a pivotal role in regulating the sensitivity of the energy mapping. It controls the saturation point of the hyperbolic tangent function, effectively distinguishing between low-amplitude sensor noise and high-amplitude semantic edges. Figure 6 illustrates the impact of varying $\tau$ on localization performance on the validation set.

We observe a convex performance trajectory. When $\tau$ is too small, the operator becomes over-sensitive to the natural thermal noise inherent in authentic regions, leading to a noisy LAD map and increased False Positives. Conversely, as $\tau$ increases, the non-linearity suppresses the subtle spectral anomalies required to identify diffusion artifacts, causing a decline in recall. The performance peaks at $\tau = 0.004$, striking an optimal balance between noise suppression and artifact retention. Theoretically, this value aligns with the pixel quantization limit ($1/255 \approx 0.0039$), suggesting that this threshold effectively suppresses quantization noise while preserving the structural reconstruction differences indicative of forgery. Consequently, we fix $\tau = 0.004$ for our experiments.

### E.2. Robustness on Image-level Detection

*Table 6.* Quantitative comparison results of image-level detection under different perturbations.

| Perturbation | MagicBrush | | SID | | CoCoGLIDE | | AutoSplice | | NanoBanana | | Qwen-Image-Edit | | Flux Kontext | | Average | |
|---|---|---|---|---|---|---|---|---|---|---|---|---|---|---|---|---|
| | Acc↑ | AP↑ | Acc↑ | AP↑ | Acc↑ | AP↑ | Acc↑ | AP↑ | Acc↑ | AP↑ | Acc↑ | AP↑ | Acc↑ | AP↑ | Acc↑ | AP↑ |
| JPEG | 0.541 | 0.612 | 0.589 | 0.688 | 0.542 | 0.587 | 0.509 | 0.533 | 0.486 | 0.497 | 0.552 | 0.583 | 0.557 | 0.592 | 0.539 | 0.585 |
| Gaussian Blur | 0.893 | 0.906 | 0.920 | 0.938 | 0.715 | 0.770 | 0.699 | 0.737 | 0.682 | 0.724 | 0.743 | 0.771 | 0.715 | 0.740 | 0.767 | 0.798 |
| Gaussian Noise | 0.592 | 0.629 | 0.621 | 0.702 | 0.561 | 0.599 | 0.542 | 0.580 | 0.521 | 0.549 | 0.545 | 0.576 | 0.564 | 0.593 | 0.564 | 0.604 |
| No Perturbation | **0.921** | **0.937** | **0.946** | **0.952** | **0.732** | **0.789** | **0.714** | **0.763** | **0.715** | **0.747** | **0.781** | **0.803** | **0.754** | **0.781** | **0.795** | **0.825** |

Table 6 details the image-level forgery detection performance under various perturbations. Consistent with the pixel-level localization results in the main paper, the detection metrics exhibit specific sensitivities to signal degradation types.

**High-Frequency Sensitivity.** The model experiences the most notable performance drop under JPEG compression and Gaussian Noise. This aligns with our theoretical premise that the LAD map relies on high-frequency spectral discrepancies (Theorem 3.3). JPEG compression quantizes high-frequency DCT coefficients, and Gaussian Noise injects global variance that masks the subtle energy gap between real and generated regions.

**Resilience to Blurring.** Interestingly, FLAME demonstrates stronger resilience to Gaussian Blur compared to noise injection. This suggests that while the finest spectral traces are attenuated by blurring, the structural

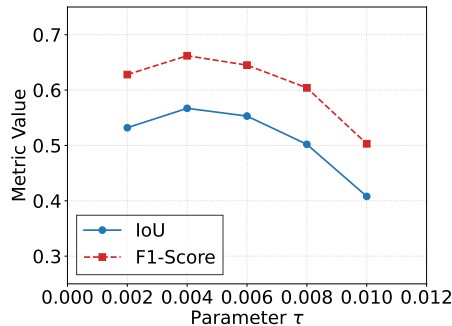

*Figure 6.* Sensitivity to threshold $\tau$.

boundary spike remains a sufficiently robust feature for the LAD-Net to discriminate between tampered and authentic images. Even under these challenging conditions, FLAME maintains an average AP significantly better than random guessing, underscoring its practical utility.

### E.3. Performance under Varying Mask Ratios

*Table 7.* Quantitative results under different mask ratios.

| Mask Ratio | IoU↑ | F1↑ |
|---|---|---|
| Small | 0.381 | 0.458 |
| Medium | 0.612 | 0.675 |
| Large | 0.742 | 0.794 |
| Overall | 0.567 | 0.652 |

To evaluate the model's sensitivity to the scale of manipulation, we stratify the test samples into three categories based on the ratio of the forged area to the total image size: Small ($< 5\%$), Medium ($5\% - 30\%$), and Large ($> 30\%$). Table 7 summarizes the results.

As anticipated in semantic segmentation tasks, performance correlates positively with object size. Large manipulations yield the highest IoU due to the abundance of statistical evidence and stronger semantic guidance for the SAM module. However, FLAME maintains a commendable IoU of $0.381$ even on Small forgeries. This indicates that the prompt-guided decoding mechanism effectively leverages the coarse cues from the LAD map to localize minute alterations, mitigating the issue where small objects are often lost during downsampling in standard CNN architectures.

### E.4. Performance across Different Editing Types

We further dissect the model's performance across the four editing operations defined in the EditStream pipeline: Object Addition, Removal, Replacement, and Background Alteration. Table 8 presents the comparative analysis on the Qwen and Flux benchmarks.

We observe that *Object Replacement* and *Addition* consistently yield higher detection scores compared to *Object Removal*. We hypothesize this stems from spectral disparity: inserting a foreign object or replacing an existing one introduces a distinct latent representation that often clashes spectrally with the original scene. In contrast, *Object Removal* typically involves inpainting background textures (e.g., extending grass or walls). The generated background texture tends to be spectrally more similar to the surrounding authentic context, resulting in a subtler energy gap and making localization more challenging.

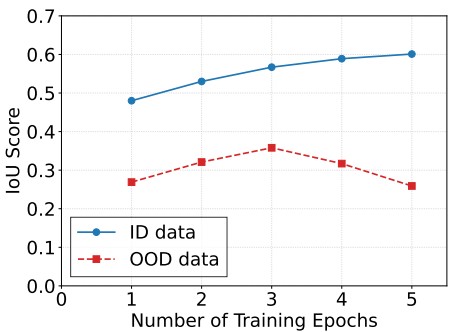

*Figure 7.* Ablation study on training epochs.

### E.5. Ablation Study on Training Epochs

Finally, we investigate the impact of training duration on generalization. Figure 7 plots the IoU scores for ID and OOD datasets across training epochs.

The results highlight a classic overfitting-to-distribution phenomenon. While performance on ID data (MagicBrush and SID) continues to improve monotonically with longer training, the zero-shot OOD performance (averaged across unseen generative models) peaks at Epoch 3 and subsequently degrades. This suggests that prolonged training causes the model to overfit to the specific artifact signatures of the training generators, reducing its ability to generalize to novel architectures like Flux or Qwen. Based on this observation, we employ an early stopping strategy at Epoch 3 to maximize cross-model robustness.

## F. Detailed Architecture of FLAME Components

In this section, we provide a granular description of the two core trainable components within the FLAME framework: the LAD-Net backbone and the Feature Adapter for SAM.

*Table 8.* Performance comparison across different editing types.

| Dataset | Object Addition | | Object Removal | | Object Replacement | | Background Alteration | |
|---|---|---|---|---|---|---|---|---|
| | IoU↑ | F1↑ | IoU↑ | F1↑ | IoU↑ | F1↑ | IoU↑ | F1↑ |
| Qwen | 0.367 | 0.429 | 0.187 | 0.263 | 0.405 | 0.463 | 0.258 | 0.317 |
| Flux | 0.307 | 0.375 | 0.165 | 0.228 | 0.371 | 0.420 | 0.261 | 0.327 |

### F.1. LAD-Net

The LAD-Net is designed to act as a specialized forensic feature extractor. Based on the implementation, the architecture consists of two key component: the Feature Encoder, and the Learnable Mask Compressor.

**Deep Separable Block (DSBlock).** The core building block of the LAD-Net feature extractor is the DSBlock, designed to maximize the receptive field while maintaining parameter efficiency. As shown in the code, the block is composed of a specific sequence of layers:

1. **Dilated Convolution:** A $3 \times 3$ dilated convolution (dilation rates increase with depth) is used to capture long-range statistical dependencies without reducing resolution.

2. **Instance Normalization (IN):** A critical design choice in our architecture is the replacement of standard Batch Normalization with Instance Normalization. Since forensic artifacts are often image-specific, Batch Normalization tends to overfit to the training domain's statistics. IN operates on individual samples, significantly enhancing the model's OOD robustness against unseen generative architectures.

3. **Squeeze-and-Excitation (SE):** We incorporate SE blocks with a reduction ratio of 8. This mechanism adaptively recalibrates channel-wise feature responses, allowing the network to suppress channels carrying irrelevant semantic information and emphasize those rich in forensic traces.

**Learnable Mask Compressor.** To generate the coarse mask $M_{coarse}$ that serves as the spatial prompt for SAM 3, we avoid simple mean pooling. Instead, we employ a lightweight sub-network consisting of a projection convolution ($1 \times 1$), followed by Instance Normalization, ReLU activation, and a final prediction convolution. This allows the model to learn an optimal non-linear projection from the high-dimensional forensic feature space to a single-channel probability map.

### F.2. Feature Adapter Module

The Feature Adapter bridges the gap between the forensic domain and the semantic domain. To preserve the powerful segmentation priors of the pre-trained SAM, we design the adapter as a residual module that injects forensic cues without disrupting the pre-trained semantic features.

Let $F_{sem} \in \mathbb{R}^{C \times H \times W}$ be the semantic features from the SAM encoder, and $F_{tex}$ be the texture features from LAD-Net. The adaptation process follows a bottleneck design:

1. **Feature Alignment:** The forensic features $F_{tex}$ are spatially interpolated to match the resolution of $F_{sem}$.

2. **Cross-Modal Fusion:** We concatenate the unadapted semantic features with the aligned forensic features. A fusion MLP ($1 \times 1$ Conv) compresses this concatenated representation into a hidden dimension $D_{hidden}$.

3. **Residual Injection:** The fused features pass through a non-linear bottleneck before being projected back to the original semantic channel dimension $C$. This produces a residual delta $\Delta F$.

The final adapted feature map $F_{adapted}$ is obtained via:

$$F_{adapted} = F_{sem} + \text{Adapter}([F_{sem} \parallel \text{Resize}(F_{tex})]) \tag{10}$$

We initialize the adapter weights using a Xavier uniform distribution to ensure stability during the early stages of fine-tuning. This residual formulation allows the gradient to flow directly through the semantic branch, ensuring that the forensic injection refines rather than overrides the object segmentation capabilities of SAM.

# G. Complete Theoretical Analysis for Artifact Separability

This appendix expands Section 3 in the main paper by (1) formalizing the full latent-diffusion inpainting pipeline with explicit equations, and (2) providing a complete, step-wise theoretical justification for artifact separability and the design of the LAD map.

## G.1. Preliminaries: Masked Latent Diffusion for Local Image Editing

**Notation.** Let $x \in \mathbb{R}^{H \times W \times 3}$ be an RGB image defined on a pixel lattice $\Omega := \{1, \ldots, H\} \times \{1, \ldots, W\}$. Let $m \in \{0, 1\}^{H \times W}$ be a binary mask, where $m(p) = 1$ indicates pixels to be edited (tampered region) and $m(p) = 0$ indicates pixels to be preserved (authentic context). We denote the edited set by $\mathcal{M} := \{p \in \Omega : m(p) = 1\}$ and the preserved set by $\mathcal{U} := \Omega \setminus \mathcal{M}$.

We consider a LDM (Ho et al., 2020; Rombach et al., 2022) consisting of: (i) a VAE encoder $E_\phi$ and decoder $D_\psi$, (ii) a time-conditional denoiser $\varepsilon_\theta(\cdot, t, c)$, optionally conditioned on text $c$.

Let the latent resolution be $h \times w$ with channel dimension $c_z$ and downsampling factor $f$. We use $\odot$ for element-wise products. We downsample the mask to the latent resolution by a deterministic operator $\mathcal{S}_f$:

$$m^{(z)} := \mathcal{S}_f(m) \in [0, 1]^{h \times w}. \tag{11}$$

For clarity of analysis we treat $m^{(z)}$ as binary mask. Soft masks can be handled by the same derivations with minor modifications.

**Step 1: VAE encoding.** The VAE defines an approximate posterior $q_\phi(z|x)$; for most practical pipelines, the latent used for editing is the posterior mean:

$$z_0^{\text{ref}} := E_\phi(x) \in \mathbb{R}^{h \times w \times c_z}. \tag{12}$$

The mapping $E_\phi$ compresses spatial information, which inevitably reduces high-frequency components.

**Step 2: Multi-step denoising with hard latent replacement.** We use the standard Gaussian forward diffusion in latent space:

$$q(z_t | z_{t-1}) = \mathcal{N}(\sqrt{\alpha_t}\, z_{t-1}, (1 - \alpha_t)I), \tag{13}$$

$$\bar{\alpha}_t := \prod_{s=1}^{t} \alpha_s, \qquad q(z_t | z_0) = \mathcal{N}(\sqrt{\bar{\alpha}_t}\, z_0, (1 - \bar{\alpha}_t)I), \tag{14}$$

$$z_t = \sqrt{\bar{\alpha}_t} z_0 + \sqrt{1 - \bar{\alpha}_t}\, \epsilon, \qquad \epsilon \sim \mathcal{N}(0, I). \tag{15}$$

The denoiser is trained with the simplified noise-prediction loss (Ho et al., 2020):

$$\mathcal{L}_{\text{LDM}} := \mathbb{E}_{z_0, t, \epsilon}\big[\|\epsilon - \varepsilon_\theta(z_t, t, c)\|_2^2\big]. \tag{16}$$

During sampling, a common DDPM-style reverse update is:

$$\hat{z}_0(z_t) := \frac{1}{\sqrt{\bar{\alpha}_t}}\Big(z_t - \sqrt{1 - \bar{\alpha}_t}\, \varepsilon_\theta(z_t, t, c)\Big), \tag{17}$$

$$\mu_\theta(z_t, t, c) := \frac{1}{\sqrt{\alpha_t}}\Big(z_t - \frac{1 - \alpha_t}{\sqrt{1 - \bar{\alpha}_t}}\, \varepsilon_\theta(z_t, t, c)\Big), \tag{18}$$

$$z_{t-1}^{\text{gen}} := \mu_\theta(z_t, t, c) + \sigma_t \eta, \qquad \eta \sim \mathcal{N}(0, I), \tag{19}$$

where $\sigma_t$ depends on the chosen sampler (DDPM/DDIM/PLMS) and can be set to 0 in deterministic sampling.

*Masked replacement (the key inpainting constraint).* To preserve the authentic region, we force the unmasked latent to match the original image latent *at every reverse step*. Specifically, we sample a reference latent at the same noise level from the forward process of the original image:

$$z_{t-1}^{\text{ref}} \sim q(z_{t-1} \mid z_0^{\text{ref}}), \qquad \text{equivalently} \qquad z_{t-1}^{\text{ref}} = \sqrt{\bar{\alpha}_{t-1}}\, z_0^{\text{ref}} + \sqrt{1 - \bar{\alpha}_{t-1}}\, \epsilon_{t-1}. \tag{20}$$

Then we composite:

$$z_{t-1} := m^{(z)} \odot z_{t-1}^{\text{gen}} \; + \; (1 - m^{(z)}) \odot z_{t-1}^{\text{ref}}. \tag{21}$$

This procedure is consistent with diffusion-based inpainting via conditioning on known pixels/regions (Lugmayr et al., 2022) and is widely adopted in latent inpainting pipelines.

**Step 3: VAE decoding.** After $T$ reverse steps, we obtain $z_0$ and decode to the edited image:

$$\hat{x} := D_\psi(z_0) \in \mathbb{R}^{H \times W \times 3}. \tag{22}$$

Note that (21) is performed in *latent space*; because VAE latents are generally not pixel-local, latent compositing is not strictly equivalent to pixel-space compositing and can introduce seam artifacts near $\partial\mathcal{M}$ (formalized later).

**Algorithm (masked latent diffusion sampling).** For completeness, the full inpainting sampler is summarized below.

---

**Algorithm 1** Masked Latent Diffusion Inpainting

---

**Input:** image $x$, mask $m$, condition $c$, diffusion steps $T$
1: $z_0^{\text{ref}} \leftarrow E_\phi(x)$; $m^{(z)} \leftarrow \mathcal{S}_f(m)$
2: Initialize $z_T \sim \mathcal{N}(0, I)$
3: **for** $t = T, \dots, 1$ **do**
3.1: $z_{t-1}^{\text{gen}} \leftarrow$ reverse update using $\varepsilon_\theta(\cdot, t, c)$
3.2: $z_{t-1}^{\text{ref}} \leftarrow q(z_{t-1} \,|\, z_0^{\text{ref}})$
3.3: $z_{t-1} \leftarrow m^{(z)} \odot z_{t-1}^{\text{gen}} + (1 - m^{(z)}) \odot z_{t-1}^{\text{ref}}$
4: $\hat{x} \leftarrow D_\psi(z_0)$
**Output:** edited image $\hat{x}$

---

**Remark.** The subsequent theory is *agnostic* to the exact sampler (DDPM vs. DDIM) as long as (1) the reverse chain is implemented with a learned denoiser and (2) hard replacement (21) is enforced.

### G.2. Artifact Separability via Gibbs Energy: Complete Analysis and Proofs

We now provide a rigorous justification for three claims:

1. **Inevitable artifacts:** diffusion-based generative pipelines inevitably introduce statistical artifacts relative to physical imaging.

2. **Inpainting amplification:** in masked inpainting, artifacts in the generated region $\mathcal{M}$ are significantly stronger than in the preserved region $\mathcal{U}$, and a sharp boundary spike emerges near $\partial\mathcal{M}$.

3. **Local separability:** these artifacts manifest as anomalies in *adjacent-pixel relations*; thus local neighbor differences yield discriminative features for real/fake separation, motivating LAD.

#### G.2.1. ENERGY MODEL ON THE IMAGE LATTICE

Following the main paper, we model the image lattice as a Gibbs Random Field (GRF) (Derin & Elliott, 1987). Let $I(p) \in \mathbb{R}^3$ denote the RGB vector at pixel $p \in \Omega$. Let $\mathcal{N}_p$ be a fixed neighborhood system (e.g., 4-neighborhood). Define a pairwise potential

$$V(p, q) := \rho(\|I(p) - I(q)\|_2), \qquad (p, q) \in \Omega \times \Omega, \; q \in \mathcal{N}_p, \tag{23}$$

where $\rho : \mathbb{R}_{\geq 0} \to \mathbb{R}_{\geq 0}$ is monotone increasing. A canonical choice for analysis is $\rho(r) = r^2$; the LAD map later uses a robust bounded surrogate via $\tanh(\cdot)$.

Define the local energy at a pixel as

$$E_{\text{loc}}(p) := \frac{1}{|\mathcal{N}_p|} \sum_{q \in \mathcal{N}_p} \|I(p) - I(q)\|_2^2. \tag{24}$$

For any set $\mathcal{S} \subseteq \Omega$, define the average local energy

$$E_{\text{loc}}(\mathcal{S}) := \mathbb{E}_{p \sim \text{Unif}(\mathcal{S})}\big[E_{\text{loc}}(p)\big]. \tag{25}$$

G.2.2. Physical imaging induces an irreducible local energy floor

We formalize the "physical entropy" argument by a standard sensor noise model.

**Assumption G.1** (Additive sensor noise floor). An authentic captured image $x$ can be locally modeled as

$$I(p) = S(p) + N(p), \tag{26}$$

where $S(p) \in \mathbb{R}^3$ is the clean irradiance-dependent signal, and $N(p)$ is stochastic sensor noise with: (i) $\mathbb{E}[N(p)] = 0$, (ii) $\mathrm{Cov}(N(p)) = \sigma^2 I_3$ with $\sigma^2 > 0$, (iii) $N(p)$ and $N(q)$ are independent for $p \neq q$ (pixel-wise temporal noise). This captures the commonly used Poisson–Gaussian approximation of shot + read noise.

**Lemma G.2** (Irreducible neighbor-difference energy). *Under Assumption G.1, for any adjacent $(p, q)$,*

$$\mathbb{E}\big[\|I(p) - I(q)\|_2^2 \mid S\big] = \|S(p) - S(q)\|_2^2 + 2 \cdot 3\sigma^2 \geq \xi_{\text{noise}}, \qquad \xi_{\text{noise}} := 6\sigma^2. \tag{27}$$

*Consequently,* $\mathbb{E}[E_{\text{loc}}(p) \mid S] \geq \xi_{\text{noise}}$ *for all* $p$.

*Proof.* Write $\Delta_S := S(p) - S(q)$ and $\Delta_N := N(p) - N(q)$, so $I(p) - I(q) = \Delta_S + \Delta_N$. Then

$$\mathbb{E}\big[\|\Delta_S + \Delta_N\|_2^2 \mid S\big] = \|\Delta_S\|_2^2 + 2\langle \Delta_S, \mathbb{E}[\Delta_N]\rangle + \mathbb{E}\big[\|\Delta_N\|_2^2\big].$$

The cross term vanishes since $\mathbb{E}[\Delta_N] = 0$. Moreover, by independence and $\mathrm{Cov}(N(p)) = \sigma^2 I_3$,

$$\mathbb{E}\big[\|\Delta_N\|_2^2\big] = \mathbb{E}\big[\|N(p)\|_2^2\big] + \mathbb{E}\big[\|N(q)\|_2^2\big] = \mathrm{tr}(\sigma^2 I_3) + \mathrm{tr}(\sigma^2 I_3) = 6\sigma^2.$$

Averaging over $q \in \mathcal{N}_p$ yields the same lower bound for $E_{\text{loc}}(p)$. $\square$

Lemma G.2 provides a concrete sufficient condition for a strictly positive energy floor in authentic regions, consistent with Assumption 3.1 in the main paper.

G.2.3. Where diffusion artifacts come from: a three-stage mechanism

We now analyze how each of the three steps in Appendix G.1 contributes to artifacts. Throughout, we use the term *artifact* to mean a systematic deviation of local statistics (e.g., $E_{\text{loc}}$) from those induced by physical imaging (Lemma G.2).

**Stage A (VAE compression): high-frequency attenuation.** A VAE encoder/decoder pair with spatial downsampling factor $f > 1$ cannot preserve all high-frequency components at the pixel scale. This is not merely an implementation detail: the compression implies that fine-scale residuals are partially projected out.

To state this formally, we adopt a standard local linearization of the VAE reconstruction operator. Let $P := D_\psi \circ E_\phi$ denote the (deterministic) VAE round-trip reconstruction map in image space.

**Assumption G.3** (Local smoothing property of VAE reconstruction). For small perturbations $\delta x$ (e.g., sensor noise), the VAE reconstruction satisfies

$$P(x + \delta x) - P(x) \approx \mathcal{H}(\delta x), \tag{28}$$

where $\mathcal{H}$ is a *non-trivial local smoothing operator* (not the identity), i.e., it mixes information within a neighborhood and attenuates pixel-wise independent components. Concretely, in a linear filter model, $\mathcal{H}$ is a convolution with a kernel $h$ having support size $\geq 2$.

**Lemma G.4** (VAE reduces the sensor-noise-induced neighbor energy). *Under Assumption G.1 and Assumption G.3, there exists a constant $\kappa_{\text{vae}} \in (0, 1)$ such that for any adjacent $(p, q)$,*

$$\mathbb{E}\big[\|P(I)(p) - P(I)(q)\|_2^2 \mid S\big] \leq \|P(S)(p) - P(S)(q)\|_2^2 + \kappa_{\text{vae}} \cdot \xi_{\text{noise}}. \tag{29}$$

*In particular, the noise contribution to local energy is* strictly attenuated *by VAE compression.*

*Proof.* Under the linearized model, the noise component after reconstruction is $\mathcal{H}(N)$. If $\mathcal{H}$ is a non-trivial local smoother (kernel support $\geq 2$), then $\mathcal{H}(N)$ becomes spatially correlated and its per-pixel variance decreases relative to i.i.d. noise (a standard property of averaging filters). For a convolutional $\mathcal{H}$, the neighbor difference $\mathcal{H}(N)(p) - \mathcal{H}(N)(q)$ is a linear combination of i.i.d. terms with coefficients given by differences of shifted kernels; the resulting variance is strictly smaller than that of $N(p) - N(q)$ unless $\mathcal{H}$ is identity. Thus there exists $\kappa_{\text{vae}} \in (0, 1)$ (depending on $h$ and the neighbor offset) such that $\mathbb{E}\|\mathcal{H}(N)(p) - \mathcal{H}(N)(q)\|_2^2 \leq \kappa_{\text{vae}}\mathbb{E}\|N(p) - N(q)\|_2^2 = \kappa_{\text{vae}}\xi_{\text{noise}}$. Adding the deterministic signal term yields (29). $\square$

**Stage B (denoiser/sampler): variance shrinkage and spectral bias.** Diffusion denoisers are trained under an $\ell_2$ objective in Eq. (main Eq. (1)). Even in classical estimation theory, the $\ell_2$ optimal predictor is a conditional mean (an $L^2$ projection), which *reduces variance* relative to the raw stochastic signal. Moreover, neural networks are empirically and theoretically known to prioritize low-frequency patterns, which implies slower/poorer fitting of high-frequency stochastic components.

We use the following formal abstraction:

**Assumption G.5** (High-frequency variance deficit of diffusion synthesis). Let $\hat{x}_{\text{gen}}$ be a region generated by the diffusion sampler and decoded by $D_\psi$. There exists $\kappa_{\text{diff}} \in (0, 1)$ such that for adjacent $(p, q)$ inside the generated region,

$$\mathbb{E}\big[\|\hat{x}_{\text{gen}}(p) - \hat{x}_{\text{gen}}(q)\|_2^2\big] \leq \mathbb{E}\big[\|P(I)(p) - P(I)(q)\|_2^2\big] - (1 - \kappa_{\text{diff}})\xi_{\text{noise}}. \tag{30}$$

Intuitively, the generated region has *less* fine-scale stochastic variation than even the VAE-reconstructed authentic content.

Assumption G.5 is supported by: (i) $\ell_2$-trained denoisers behaving like conditional mean estimators on weak-SNR components, and (ii) spectral bias results showing that high-frequency modes are learned slower and remain underrepresented under finite training/sampling.

**Stage C (hard latent replacement): seam discontinuity and covariance mismatch.** Finally, the key inpainting operation (21) stitches two latent random fields:

$$z_{t-1}^{\text{gen}} \quad \text{(model sample)} \qquad \text{and} \qquad z_{t-1}^{\text{ref}} \quad \text{(noised reference latent)}.$$

Even if both are plausible latents, they are *not* drawn from the same conditional distribution at the boundary, and hard replacement creates a piece-wise latent manifold. This violates the smoothness/coherence prior of natural images and yields seam artifacts after decoding.

We formalize this via a boundary-crossing energy lower bound.

**Lemma G.6** (Cross-boundary energy is strictly larger). *Let $p \in \mathcal{U}$ and $q \in \mathcal{M}$ be adjacent across the boundary. Assume that (conditionally on coarse content) the preserved pixel $I_\mathcal{U}(p)$ and generated pixel $I_\mathcal{M}(q)$ have different second-order statistics:*

$$\mathbb{E}[I_\mathcal{U}(p)] \neq \mathbb{E}[I_\mathcal{M}(q)] \quad or \quad \text{Cov}(I_\mathcal{U}(p)) \neq \text{Cov}(I_\mathcal{M}(q)). \tag{31}$$

*Then the expected cross-boundary squared difference satisfies*

$$\mathbb{E}\big[\|I_\mathcal{U}(p) - I_\mathcal{M}(q)\|_2^2\big] \geq \mathbb{E}\big[\|I_\mathcal{U}(p) - I_\mathcal{U}(q')\|_2^2\big] + \Delta_\partial, \tag{32}$$

*for some $\Delta_\partial > 0$ and any $q' \in \mathcal{N}_p \cap \mathcal{U}$, provided the mismatch in (31) is non-degenerate.*

*Proof.* Let $X := I_\mathcal{U}(p)$ and $Y := I_\mathcal{M}(q)$. Then

$$\mathbb{E}\|X - Y\|_2^2 = \|\mathbb{E}X - \mathbb{E}Y\|_2^2 + \text{tr}\big(\text{Cov}(X) + \text{Cov}(Y) - 2\text{Cov}(X, Y)\big).$$

Across the boundary, the dependence between $X$ and $Y$ is weaker than within a single coherent field, so $\text{Cov}(X, Y)$ cannot fully cancel the sum of marginal covariances. Moreover, any mean mismatch contributes the nonnegative term $\|\mathbb{E}X - \mathbb{E}Y\|_2^2$. In contrast, within $\mathcal{U}$, for $Y' := I_\mathcal{U}(q')$, the pair $(X, Y')$ shares matched statistics and stronger local correlation, yielding smaller expected squared differences. Thus there exists $\Delta_\partial > 0$ whenever the mismatch (31) is non-degenerate. □

Lemma G.6 captures the *boundary spike* phenomenon; it is further amplified in latent pipelines because latent compositing is not pixel-equivalent.

G.2.4. MAIN RESULTS: INEVITABILITY, INPAINTING AMPLIFICATION, AND LOCAL SEPARABILITY

We now prove the three target claims.

**(1) Diffusion models inevitably introduce artifacts.** We formalize this as an energy-statistic separation between physically captured images and LDM-generated images.

**Theorem G.7** (Unavoidable energy anomaly of latent diffusion synthesis). *Assume the authentic imaging process satisfies Assumption G.1. Consider any edited image $\hat{x}$ produced by the three-stage pipeline in Appendix G.1. If (i) the VAE reconstruction attenuates pixel-wise independent noise (Lemma G.4) and (ii) the diffusion synthesis exhibits a high-frequency variance deficit (Assumption G.5), then there exists $\delta > 0$ such that*

$$E_{\mathrm{loc}}(\mathcal{M}) \leq E_{\mathrm{loc}}(\mathcal{U}) - \delta. \tag{33}$$

*Equivalently, the generated region necessarily differs from the preserved region in the local neighbor-difference energy, hence artifacts are unavoidable under this statistic.*

*Proof.* Inside the preserved region $\mathcal{U}$, the masked replacement (21) forces $z_t^{\mathcal{U}}$ to follow the forward noised reference latent at every step; thus the decoded output in $\mathcal{U}$ is effectively anchored to the VAE reconstruction of the original content, up to small decoding imperfections. Therefore, by Lemma G.4, adjacent differences in $\mathcal{U}$ retain a nontrivial fraction of the physical noise-induced energy.

Inside the generated region $\mathcal{M}$, the latent is produced by the learned reverse process and then decoded. By Assumption G.5, the expected adjacent squared differences in $\mathcal{M}$ are strictly smaller than those in the VAE-reconstructed authentic content by a margin proportional to $\xi_{\mathrm{noise}}$. Averaging over neighbors and over pixels in $\mathcal{U}$ and $\mathcal{M}$ yields the claimed strict gap with $\delta := (1 - \kappa_{\mathrm{diff}})\xi_{\mathrm{noise}}$ (up to constant factors absorbed by averaging). $\square$

**(2) Inpainting causes larger artifacts in the edited region and a boundary spike.** We now give a complete proof of the energy gap and boundary spike stated as Theorem 3.3 in the main paper, with explicit dependence on the three-stage mechanism.

**Theorem G.8** (Energy Gap & Boundary Spike (full proof)). *Let $\hat{x}$ be produced by masked latent diffusion inpainting (Algorithm 1). Assume Assumption G.1, Assumption G.3, and Assumption G.5. Then:*

1. *__Interior Gap:__ $E_{\mathrm{loc}}(\mathcal{U}) > E_{\mathrm{loc}}(\mathcal{M})$.*

2. *__Boundary Spike:__ Let $\partial\mathcal{M} := \{p \in \Omega : \exists q \in \mathcal{N}_p \text{ s.t. } m(p) \neq m(q)\}$. Then $E_{\mathrm{loc}}(\partial\mathcal{M}) \gg E_{\mathrm{loc}}(\mathcal{M})$ in the sense that*

$$E_{\mathrm{loc}}(\partial\mathcal{M}) \geq E_{\mathrm{loc}}(\mathcal{U}) + \Delta_\partial, \tag{34}$$

*for some $\Delta_\partial > 0$ determined by the cross-boundary statistical mismatch.*

*Proof.* (**Interior gap**). Fix an adjacent pair $(p, q)$ fully inside $\mathcal{U}$. Because $z_{t-1}$ is replaced by $z_{t-1}^{\mathrm{ref}}$ on $\mathcal{U}$ for every $t$, the decoded output on $\mathcal{U}$ is anchored to $P(I)$, where $I$ is the authentic image content. Thus, for $(p, q) \subset \mathcal{U}$,

$$\mathbb{E}\|\hat{x}(p) - \hat{x}(q)\|_2^2 \approx \mathbb{E}\|P(I)(p) - P(I)(q)\|_2^2.$$

Now fix an adjacent pair $(p, q)$ fully inside $\mathcal{M}$. By Assumption G.5,

$$\mathbb{E}\|\hat{x}(p) - \hat{x}(q)\|_2^2 \leq \mathbb{E}\|P(I)(p) - P(I)(q)\|_2^2 - (1 - \kappa_{\mathrm{diff}})\xi_{\mathrm{noise}}.$$

Averaging over neighbors and pixels yields $E_{\mathrm{loc}}(\mathcal{M}) \leq E_{\mathrm{loc}}(\mathcal{U}) - \delta$ for $\delta > 0$, hence $E_{\mathrm{loc}}(\mathcal{U}) > E_{\mathrm{loc}}(\mathcal{M})$.

(**Boundary spike**). Consider a boundary pixel $p \in \partial\mathcal{M}$. By definition, at least one neighbor $q \in \mathcal{N}_p$ lies on the opposite side of the mask. Such neighbor pairs are cross-boundary pairs of the form treated in Lemma G.6. Under non-degenerate mismatch of statistics across the stitched fields (which is induced by the fact that one side is anchored to $z^{\mathrm{ref}}$ and the other is sampled from $z^{\mathrm{gen}}$ and then decoded), Lemma G.6 implies a strict increase in expected squared differences for at least one neighbor term in the sum (24). Therefore, $E_{\mathrm{loc}}(p)$ is larger than the within-$\mathcal{U}$ baseline by at least $\Delta_\partial/|\mathcal{N}_p|$ in expectation, and averaging over $p \in \partial\mathcal{M}$ yields $E_{\mathrm{loc}}(\partial\mathcal{M}) \geq E_{\mathrm{loc}}(\mathcal{M}) + \Delta_\partial$ (absorbing neighborhood constants into $\Delta_\partial$). $\square$

**(3) Local adjacency differences yield a separable artifact feature (LAD).** Theorem G.8 states that inpainting creates a characteristic topology: a low-energy hole surrounded by a high-energy ring, against a higher-energy background. We now show that comparing *adjacent pixel differences* is sufficient to expose this topology.

Recall the LAD operator (Definition 3.4 in the main paper):

$$L(p) := \frac{1}{|\mathcal{N}_p|} \sum_{q \in \mathcal{N}_p} \tanh\left(\frac{\|I(p) - I(q)\|_2^2}{\tau^2}\right), \tag{35}$$

where $\tau > 0$ is a temperature. Let $\Psi(u) := \tanh(u/\tau^2) \in (0, 1)$.

**Lemma G.9** (LAD is a robust monotone transform of local energy). *For any $p$, $L(p)$ is a bounded, monotone increasing functional of the set $\{\|I(p) - I(q)\|_2^2 : q \in \mathcal{N}_p\}$. Moreover, for any nonnegative random variable $X$, $\Psi$ is concave on $X \geq 0$ and satisfies*

$$\mathbb{E}[\Psi(X)] \leq \Psi(\mathbb{E}[X]). \tag{36}$$

*Proof.* Monotonicity and boundedness are immediate from $\tanh$ being monotone and bounded in $(0, 1)$ for positive inputs. Concavity of $\tanh$ on $\mathbb{R}_{\geq 0}$ is given by $\tanh''(u) = -2\tanh(u)\,\mathrm{sech}^2(u) < 0$ for $u > 0$. Jensen's inequality yields (36). $\square$

**Theorem G.10** (Artifact separability in LAD space). *Assume the conditions of Theorem G.8. Then there exist constants $0 < \mu_{\mathcal{M}} < \mu_{\mathcal{U}} < \mu_{\partial}$ such that*

$$\mathbb{E}[L(p) \mid p \in \mathcal{M}] \leq \mu_{\mathcal{M}}, \qquad \mathbb{E}[L(p) \mid p \in \mathcal{U}] \geq \mu_{\mathcal{U}}, \qquad \mathbb{E}[L(p) \mid p \in \partial\mathcal{M}] \geq \mu_{\partial}. \tag{37}$$

*Furthermore, because $L(p)$ averages bounded terms, it concentrates around its expectation: for any $\epsilon > 0$,*

$$\mathbb{P}\big(|L(p) - \mathbb{E}L(p)| \geq \epsilon\big) \leq 2\exp\big(-2|\mathcal{N}_p|\epsilon^2\big). \tag{38}$$

*Hence, a simple threshold on $L(p)$ can separate $\mathcal{M}$ and $\mathcal{U}$ with high probability, and pixels on $\partial\mathcal{M}$ saturate to high LAD responses.*

*Proof.* By Theorem G.8, the distribution of adjacent squared differences inside $\mathcal{M}$ is shifted to smaller values than inside $\mathcal{U}$, while cross-boundary differences are shifted to larger values. Since $\Psi$ is monotone increasing, these shifts imply an ordering of expectations of $\Psi(\|I(p) - I(q)\|_2^2)$ across regions, and averaging over neighbors preserves the ordering, yielding (37) for some constants $(\mu_{\mathcal{M}}, \mu_{\mathcal{U}}, \mu_{\partial})$.

For concentration, define $Y_q := \Psi(\|I(p) - I(q)\|_2^2) \in (0, 1)$ for each neighbor $q \in \mathcal{N}_p$. Then $L(p) = \frac{1}{|\mathcal{N}_p|}\sum_q Y_q$ is an average of bounded random variables. Hoeffding's inequality gives (38) (independence is not strictly required for a qualitative concentration statement; bounded-difference martingale variants yield similar bounds under mild dependence).

Thus LAD provides a robust, locally aggregated estimator that preserves the energy gap (interior) and emphasizes the boundary spike, enabling reliable discrimination of $\mathcal{M}$ vs. $\mathcal{U}$. $\square$

**Implication for method design.** Theorems G.7–G.10 establish a principled route from diffusion editing mechanics (VAE compression + denoiser spectral bias + hard latent stitching) to a *local adjacency* statistic that is separable for authentic vs. generated regions. This directly motivates using the LAD map as an explicit artifact representation, upon which a downstream segmentation/refinement model can operate.

# H. Hyperparameter Values

*Table 9.* Optimal hyperparameter values with sweep range.

| Hyperparameter | Optimal | Sweep Range |
|---|---|---|
| Learning rate | 0.001 | $\{0.01, 0.001, 0.0001, 0.00001\}$ |
| Temperature $\tau$ | 0.004 | $\{0.002, 0.004, 0.006, 0.008, 0.010\}$ |
| Focal $\alpha$ | 0.6 | $\{0.5, 0.55, \ldots, 0.75, 0.80\}$ |
| Focal $\gamma$ | 1.0 | $\{1.0, 1.25, \ldots, 1.75, 2.0\}$ |
| Adam weight decay | 0.0001 | $\{0.0001, 0.00001, 0.0\}$ |
| Dropout rate | 0.2 | $\{0.0, 0.1, 0.15, 0.2, 0.25, 0.3\}$ |
| Batch size | 8 | $\{2, 4, 8, 16\}$ |

# I. Prompts for EditStream

---

**Semantic Planner Instruction**

You are a highly intelligent **Semantic Planner** for an image editing dataset generation pipeline. Your task is to analyze the provided image and generate a realistic, logic-consistent editing instruction.

**Please follow these steps:**

1. **Scene Analysis:** Identify salient objects and the background environment.

2. **Feasibility Check:** Determine which editing operations are visually and semantically feasible for this specific image.

3. **Operation Selection:** Select **ONE** operation from the following categories based on feasibility (aim for diversity):

   - **Object Addition:** Insert a new object that fits the context (e.g., "Add a red car in the driveway").
   - **Object Removal:** Erase an existing entity (e.g., "Remove the pedestrian").
   - **Object Replacement:** Swap an object with a semantically different one.
   - **Background Alteration:** Modify environmental textures or styles (e.g., "Change the grass to snow").

4. **Output Generation:** Provide the edit instruction and the approximate bounding box of the target region.

---

**Output Format (JSON strictly):**

```
{
"scene_description":  "Brief description of the scene...",
"selected_operation":  "Object Replacement", // One of the 4 categories
"edit_instruction":  "Replace the brown dog with a white cat",
"target_object_name":  "brown dog",
"target_region_box":  [x_min, y_min, x_max, y_max], // Normalized [0-1]
"reasoning":  "The dog is clearly visible and replacing it creates a meaningful
semantic change."
}
```

**Autonomous Model Scouting Agent**

You are an **Autonomous Model Scouting Agent** tasked with identifying the latest and most capable image inpainting models on Hugging Face.

You have access to a tool named search_hf_models. Your mission is:

1. **Search:** Use the tool to find models with the tag image-to-image.

2. **Filter & Rank:** Analyze the search results based on the following multi-metric criteria:
   - **Popularity:** High download counts and likes.
   - **Recency:** Prioritize models updated within the last 3 months (trending status).
   - **Architecture:** Look for diverse architectures.

3. **Selection:** Select the top-3 distinct candidates that are most promising for high-quality image editing.

**Response Requirements:**

- Call the search tool first.

- After receiving results, output a list of the 3 selected model_ids.

- Provide a brief justification for each selection based on the metrics.

## Senior AI Software Engineer

You are a Senior AI Software Engineer specializing in ModelOps. I will provide you with the raw text content of a Hugging Face Model Card (README) for a specific inpainting model.

**Your Task:**

Synthesize a standardized Python wrapper class named `EditStreamWrapper` for this model.

**Requirements:**

1. **Interface Standardization:** The class must implement a `generate(self, image, mask, prompt)` method.

   - `image`: Input PIL Image (RGB).
   - `mask`: Binary PIL Image (White=Edit, Black=Keep).
   - `prompt`: String describing the edit.

2. **Constraint Parsing:** Analyze the README text to identify:

   - Resolution limits (e.g., if it requires 512x512, implement auto-resizing in the wrapper).
   - Specific input formatting (e.g., strictly masked inputs vs. original image + mask).

3. **Dependency Handling:** Include a valid `__init__` method that loads the model using `diffusers` or the specific library mentioned in the README. Use CUDA if available.

4. **Error Handling:** Add basic error handling for dimension mismatches.

**Input Model Card Content:**

```
"""
{{INSERT_MODEL_CARD_CONTENT_HERE}}
"""
```

**Output:**

Return ONLY the executable Python code block containing the `EditStreamWrapper` class.

