# OpenReview forum: "Order within Chaos: Capturing Intrinsic Energy Anomalies for AI-Manipulated Image Forgery Localization"
_ICML.cc/2026/Conference — ICML 2026 regular_

### Official Review · Reviewer_8qBo · 2026-03-02

**Soundness:** 3
**Presentation:** 3
**Significance:** 3
**Originality:** 3
**Overall Recommendation:** 4
**Confidence:** 4

**Summary:**

This paper propose FLAME, a framework of localizing AI-manipulated image forgeries, specifically those generated by diffusion-based editing models. The method introduces a Local Adjacency Discrepancy (LAD) map to quantify these energy anomalies and employs a coarse-to-fine architecture combining a lightweight LAD-Net with a parameter-efficient adapter for SAM3 to achieve pixel-level precision. Additionally, the paper presents EditStream, an automated pipeline for continuous training data synthesis. Extensive experiments demonstrate state-of-the-art performance across multiple benchmarks.

**Compliance With Llm Reviewing Policy:**

Affirmed.

**Final Justification:**

The rebuttal resolves most of my concerns, and I maintain my current recommendation.

**Key Questions For Authors:**

See weaknesses.

**Limitations:**

This paper does not discuss limitations. Could the authors provide an analysis of typical failure cases?

**Strengths And Weaknesses:**

Strengths:
1. The paper provides a mathematical analysis using Gibbs energy modeling to explain why diffusion models leave intrinsic statistical artifacts, moving beyond empirical observation to principled forensic design.
2. The proposed self-evolving data synthesis pipeline addresses the critical issue of benchmark lag by autonomously scouting and integrating new generative models, ensuring the detector remains relevant against evolving threats.
3. The method achieves state-of-the-art performance on multiple OOD benchmarks and demonstrates robustness against common perturbations like JPEG compression and noise.

Weaknesses:
1. Table 8 indicates significantly lower performance on object removal tasks compared to addition or replacement, attributed to spectral similarity with background textures, but no specific mitigation strategy is proposed for this challenging case.
2. Although the adapter is lightweight, the inference time and memory usage of the full pipeline (LAD-Net+SAM 3) are not reported, making it difficult to assess feasibility for real-time deployment.
3. The autonomous agent generates training samples without human intervention. What mechanisms are in place to filter low-quality or semantically inconsistent edits that might negatively impact model convergence?

---

> ### Author Rebuttal · Authors · 2026-03-31
>
> We sincerely thank the reviewer for the constructive feedback. We appreciate your recognition of the principled Gibbs-energy motivation, the value of the self-evolving EditStream pipeline, and the strong OOD performance and robustness of FLAME. We have added targeted discussion on object removal, deployment cost, sample filtering, and failure cases to address the concerns below.
>
> ### W1. Lower performance on object removal and possible mitigation
>
> Object removal is indeed the hardest editing type in Table 8. Compared with addition or replacement, it usually relies on inpainting background texture, so the filled region can remain statistically close to the surrounding authentic context. This weakens the LAD interior cue and makes the coarse forensic signal less separable.
>
> This difficulty also affects refinement. Addition or replacement often introduces a new object or a clearer semantic transition, which gives SAM-based correction a stronger anchor. By contrast, removal may look locally plausible and may not present a strong object-level boundary. In short, removal is harder because both the forensic cue and the semantic anchor become weaker.
>
> Three directions can address this issue: (i) removal-focused hard-case generation in EditStream, especially on texture-continuous backgrounds such as grass, walls, sky, or water; (ii) boundary-aware supervision or sampling around subtle seams; and (iii) stronger context-consistency modeling in refinement. We will add this mitigation roadmap explicitly in the revision.
>
> ### W2. Inference time and memory usage
>
> We now benchmark the full pipeline on an NVIDIA A800 80GB PCIe GPU at 512x512, batch size 1:
>
> | Component | Mean Latency (ms) | Peak Incremental Memory |
> | --- | ---: | ---: |
> | Full pipeline | 36.2 | 128.7 MB |
> | LAD / forensic branch | 9.0 | 124.0 MB |
> | SAM refinement total | 28.4 | 84.5 MB |
> | SAM encoder only | 25.9 | 84.5 MB |
> | Adapter + decoder | 3.85 | 63.0 MB |
>
> | Post-load GPU memory | Value |
> | --- | ---: |
> | Allocated | 0.368 GB |
> | Peak allocated during end-to-end inference | 526.1 MB |
>
> These numbers show that the main overhead comes from the SAM-based refinement stage, while the LAD branch itself is lightweight. Overall, the full pipeline is fast and memory-efficient enough for practical deployment, especially for server-side forensic screening and batched analysis. We will add these measurements to make the deployment cost explicit in the paper.
>
> ### W3. Filtering low-quality or semantically inconsistent synthetic samples
>
> EditStream already includes several safeguards to reduce low-quality or semantically inconsistent edits before dataset insertion:
>
> 1. the semantic planner restricts edits to semantically feasible regions and operations;
> 2. SAM-generated masks enforce boundary-aligned target regions; and
> 3. complexity-aware source selection avoids trivial or degenerate scenes.
>
> To strengthen this further, we will add a final admission-control verifier after edit generation and before dataset insertion. It will use a two-stage filter: a lightweight deterministic pre-check for mask validity and localization quality, followed by a VLM-based judge for instruction fidelity, semantic consistency, and artifact severity. Samples that fail these checks are discarded or regenerated. This complements the current planning-based safeguards with explicit sample-level verification.
>
> ### Limitation. Typical failure cases
>
> We will add an explicit Limitations paragraph and representative failure-case visualizations in the revision. The typical failures arise only when the following two conditions hold simultaneously:
>
> 1. the LAD cue is weak or non-unique, such as in large smooth authentic regions or scenes with repetitive local structures; and
> 2. the semantic refinement stage lacks a clear boundary prior, such as in very small edits or scenes with many similar candidate objects.
>
> In such cases, the coarse anomaly map becomes diffuse, and SAM 3 may not recover the correct target region. Representative anonymous examples are shown here: [failure cases](https://postimg.cc/Mn7y8s1y).

---

> > ### Author Rebuttal · Reviewer_8qBo · 2026-04-01
> >
> > Thanks for your response, my concerns have been addressed.

---

> > > ### Author Response · Authors · 2026-04-05
> > >
> > > Thank you for confirming your concerns are resolved. We would be deeply grateful if you might consider reflecting this in your final evaluation.

---

### Official Review · Reviewer_ZED5 · 2026-03-10

**Soundness:** 2
**Presentation:** 3
**Significance:** 2
**Originality:** 2
**Overall Recommendation:** 5
**Confidence:** 4

**Summary:**

This paper introduces FLAME, a pixel-level AI forgery localization framework. The core premise is that diffusion models smooth out the natural high-frequency sensor noise in real images (the authors frame as a "Gibbs energy anomaly"). To exploit this, the authors propose a Local Adjacency Discrepancy (LAD) map, process it through a lightweight LAD-Net, and feed it into a SAM 3 model via an Adapter for refined mask generation. Additionally, they introduce EditStream, an automated pipeline that scrapes HuggingFace models to synthesize training data on the fly, keeping the model up to date with the latest generative priors.

**Compliance With Llm Reviewing Policy:**

Affirmed.

**Final Justification:**

**Score Adjustment Justification**

Thank you to the author for the excellent rebuttal. I am very satisfied with the responses and the additional work provided.

The authors have clearly and completely addressed all of my previous concerns. The new explanations and experiments are very convincing.

Because my concerns are fully resolved, I am happy to increase my score from 3 to 5.

**Key Questions For Authors:**

1. Brittleness of $\tau$: How does your fixed threshold $\tau$ hold up when evaluating real images captured under extreme conditions (e.g., extremely low light with high ISO, or aggressive smartphone ISP denoising)? The natural noise profile is already destroyed there; doesn't your pipeline flag these as fakes?
2. Adaptive Attacks: What happens if an attacker simply injects camera-matched sensor noise into the diffusion-generated image before passing it to your detector? It seems the LAD map would be completely neutralized.
3. Ablation on Data vs. Architecture: EditStream provides a massive advantage in data scale and freshness. If we were to fine-tune your baseline competitors on the exact same EditStream dataset, would FLAME still maintain its margin? I suspect the performance bump comes almost entirely from the data, not the LAD-Net architecture.

**Limitations:**

The authors briefly touch upon limitations, but they significantly downplay the fragility of their method. The extreme sensitivity to the window size and the $\tau$ threshold, along with the inherent vulnerability to simple post-processing noise injection attacks, are critical flaws that need a much more transparent discussion.

**Strengths And Weaknesses:**

Strengths：
1. Engineering & Pipeline: The EditStream pipeline is a genuinely practical engineering effort. Creating a closed-loop system that continuously synthesizes data to keep pace with evolving diffusion models is a solid approach to the domain-shift problem.
2. Empirical Performance: The zero-shot boundary refinement leveraging SAM 3 yields visually pleasing masks, and the evaluations across OOD datasets and various robustness perturbations (JPEG, noise) show competitive metrics (IoU/F1).

Weaknesses：
1. Soundness: Over-engineered Core Metric: To be perfectly frank, the "Gibbs energy anomaly" is a heavily over-engineered way of describing a simple local high-pass filter. The LAD map fundamentally boils down to local pixel variance with a non-linear truncation. The heavy mathematical machinery (Eq. 3-5 and the Theorems in the appendix) feels shoehorned in to make a basic statistical difference look like a profound physical discovery. Furthermore, the hardcoded hyperparameter $\tau=0.004$ makes the method incredibly brittle to real-world ISP variations.
2. Presentation: Obscured Intuition: Because of the excessive mathematical dressing, the presentation suffers. The text spends too much time on dense equations for what is essentially "real images have sensor noise, generated ones don't," obscuring the actual simplicity of the method.
3. Significance: Vulnerability in the Wild: The practical significance is limited because the defense mechanism is structurally naive. A savvy attacker can easily bypass this by applying a post-hoc Poisson-Gaussian noise model matching real camera profiles, completely breaking the LAD map assumption.
4. Originality Incremental Assembly: The originality is marginal. Exploiting local high-frequency/noise discrepancies is a concept dating back to Noiseprint. Pairing this old concept with a modern SAM 3 adapter is a standard empirical "trick" nowadays, and the scraping agent (EditStream) heavily overlaps with recent concurrent works like FakeShield.

---

> ### Author Rebuttal · Authors · 2026-03-31
>
> We thank the reviewer for the feedback and for recognizing EditStream's practical value and FLAME's strong performance. Below we clarify the scope of our claims, add stronger controls, and make the threat model and limitations more explicit.
> ### W1. Is LAD just a local filter?
> We do not think LAD is interchangeable with a generic local high-pass / variance filter. Table 5 provides the clearest evidence: with the same 3x3 neighborhood, LAD reaches 0.567/0.662, versus 0.502/0.596 for Avg, 0.451/0.542 for Max, and 0.439/0.525 for Min. If the method were merely an arbitrary local filter, these standard alternatives should be much closer; instead, the operator choice clearly matters.
>
> The reason is that LAD is designed for a specific cue in mixed real/synthetic edits: a lower-response edited interior together with a boundary spike at the authentic/generated interface (Theorem 3.3), rather than generic local variance amplification. Eq. (5) reflects this design: pairwise differences capture local inconsistency, `tanh` suppresses dominant semantic edges, and neighborhood aggregation stabilizes a region-level response instead of a generic edge map. Our claim is therefore task-specific: LAD is a theoretically motivated operator for AI-edited forgery localization, not a broader physical law.
> ### W2. Presentation
> We agree that the practical intuition should appear before the full derivation. In the LAD map, edited regions appear as a lower-response interior with a boundary spike near the authentic/generated interface. In revision, we will lead with this intuition and the LAD-map visualization, then introduce Eq. (5), keeping the Gibbs/MRF-style derivation as supporting material.
> ### W3 & Q2. Adaptive attacks / vulnerability in the wild
> We additionally conduct an adaptive PGD attack on the LAD map as a strong proxy for detector-aware noise injection, reducing average IoU/F1 to 0.185/0.243. This is a strong attack, but not a full collapse. The reason is that it mainly damages the LAD-driven coarse cue. When some coarse signal still remains, SAM 3 can still refine it, but once the proposal is heavily degraded, refinement also drops sharply. We therefore limit robustness claims to common post-processing and OOD editor shift, and treat detector-aware anti-forensic attacks as a limitation.
> ### W4. Originality
> We appreciate this originality concern, but we do not think FLAME is merely an incremental assembly.
>
> 1. Noiseprint-like methods address traditional physical forensics through camera-residual fingerprints, which degrade on AI-edited images. FLAME instead localizes mixed real/generated edits, so LAD is designed for the resulting interior-boundary discrepancy.
> 2. The contribution is not simply adding an adapter to SAM 3, but coupling forensic and semantic cues: LAD-Net provides the proposal, and SAM 3 refines it. Table 4 shows that removing either side causes a large drop.
> 3. FakeShield-like methods are localization or verification pipelines. EditStream addresses a different limitation: static training data becomes outdated as new editors emerge, so it continuously scouts new editing models and refreshes the training distribution.
>
> Our contribution is therefore the FLAME system for AI-edited forgery localization under evolving editors.
> ### Q1. Fixed threshold $\tau$
> Appendix D.1 shows that $\tau$ is not arbitrary: performance peaks at $\tau = 0.004$, very close to the 8-bit quantization step $1/255 ≈ 0.0039$. In Eq. (5), this sets sensitivity near one quantization level, suppressing digitization-scale fluctuations while preserving the larger discrepancies caused by diffusion smoothing and real/generated mixing.
>
> Extreme low-light or aggressive ISP denoising is still a harder regime because it perturbs the sensor statistics used by LAD. The likely effect is higher false positives or noisier coarse proposals, not universal failure, because full FLAME still relies on localized interior-boundary structure and SAM 3 refinement. We will state this boundary explicitly in the revision.
> ### Q3. Data vs architecture
> To directly answer this question, and within the rebuttal-time budget, we fine-tune two representative baselines, TruFor and SIDA, on the exact same Qwen-Image-Edit subset used for FLAME-F:
>
> |Model|Qwen|Flux|OOD avg|
> |-|-|-|-|
> |TruFor|0.337/0.466|0.297/0.382|0.303/0.393|
> |SIDA|0.254/0.313|0.190/0.274|0.268/0.345|
> |FLAME-F|0.482/0.603|0.446/0.548|0.460/0.565|
>
> Under this same-data control, FLAME-F still leads on Qwen and remains clearly ahead on Flux and OOD average. If fresher data were the whole explanation, this margin should largely disappear; it does not. This is also consistent with Table 4, where removing the LAD cue or the forensic-semantic coupling causes large drops.
> ### Limitations
> We will add an explicit Limitations paragraph. The main boundaries are: weak or non-unique artifact cues, ambiguous semantics, stronger degradation under heavy compression and global noise, and detector-aware anti-forensic attacks.

---

> > ### Author Rebuttal · Reviewer_ZED5 · 2026-04-03
> >
> > **Score Adjustment Justification**
> >
> > Thank you to the author for the excellent rebuttal. I am very satisfied with the responses and the additional work provided.
> >
> > The authors have clearly and completely addressed all of my previous concerns. The new explanations and experiments are very convincing.
> >
> > Because my concerns are fully resolved, I am happy to increase my score from 3 to 5.

---

### Official Review · Reviewer_FDbh · 2026-03-11

**Soundness:** 2
**Presentation:** 3
**Significance:** 3
**Originality:** 2
**Overall Recommendation:** 4
**Confidence:** 4

**Summary:**

This research proposes an innovative forensic perspective, arguing that while AI-generated images lack the physical noise characteristic of traditional sensors, their diffusion-based generation process inherently suppresses local high-frequency variance. This creates an "artificial order" or low-energy anomaly that is statistically distinct from the "natural chaos" of authentic images. Based on this theoretical insight, the authors developed FLAME, a coarse-to-fine localization framework. At its core, the framework utilizes Local Adjacency Discrepancy (LAD) maps to precisely capture these energy anomalies, combining a lightweight LAD-Net for initial mask generation with a SAM3-based semantic refinement module to achieve pixel-level precision. To address the issue of forensic benchmarks lagging behind the rapid evolution of generative technologies, the study also introduces EditStream, an automated multi-agent pipeline capable of continuous, instruction-based synthesis of high-quality training data. Extensive experimental results demonstrate that FLAME achieves state-of-the-art localization accuracy across several mainstream benchmarks and exhibits superior generalization and robustness to unseen generative models.

**Compliance With Llm Reviewing Policy:**

Affirmed.

**Key Questions For Authors:**

(1) Since the "natural entropy" of naturally smooth or low-texture areas is inherently low and the energy difference between them and AI-generated content might be extremely small, how does the framework distinguish normal low-variance regions in real images from suppressed-variance regions in forgeries, and do specific false-positive cases exist in such areas?
(2) How does the LAD map's discriminative power perform under various levels of JPEG compression?
(3) The authors should provide a more detailed ablation study comparing the performance of LAD-Net alone with the full FLAME framework, specifically clarifying what proportion of the localization accuracy is attributed to the "energy anomaly" features versus the "semantic priors" of SAM 3?

**Limitations:**

Yes

**Strengths And Weaknesses:**

Strengths：
(1) This research provides a rigorous foundation by exploring the suppression of local high-frequency variance in diffusion processes and mathematically defining the statistical energy discrepancy between "artificial order" and "natural chaos".
(2) The proposed FLAME framework employs an ingenious "coarse-to-fine" two-stage strategy. This design leverages both the theoretical advantages of the LAD Map and the semantic understanding of large foundation models, achieving high precision while maintaining efficiency.
(3) The paper identifies a key bottleneck in current forensic research—static benchmark datasets cannot keep pace with rapidly evolving generative models. To address this, it proposes EditStream, an automated, continuously evolving data synthesis pipeline.

Weaknesses：
(1) The core theory and method rely on the discrepancy in high-frequency statistical properties between diffusion-generated and authentic regions, which suggests that the forensic features FLAME relies on may be vulnerable to adversarial attacks aimed at removing high-frequency traces or common network transmission compression.
(2) The proposed framework invoks SAM 3's image encoder for semantic feature extraction, performing feature adaptation and fusion, and finally generating the fine mask via SAM 3's decoder. The paper does not evaluate or discuss the overall inference speed, memory footprint, or real-time processing capability of the framework, which are important factors for consideration in practical deployment scenarios.

---

> ### Author Rebuttal · Authors · 2026-03-31
>
> We sincerely thank the reviewer for the constructive feedback. We appreciate your recognition of the theoretical motivation behind LAD, the effectiveness of FLAME's design, and the importance of EditStream for evolving forensic threats. We add stronger robustness analysis, efficiency measurements, and clearer ablations below.
>
> ### W1 & Q2. Robustness
>
> We agree that both removing fine high-frequency traces and common network transmission compression weaken the forensic signal. Our claim is not that FLAME is invariant to these operations, but that under common transmission distortions the cue does not immediately collapse. This is because FLAME relies not only on the finest traces, but also on the interior energy gap and boundary mismatch, and the refine module can still correct the mask when some coarse cue remains.
>
> The original paper already reports robustness under JPEG-75 and Gaussian Noise in Table 3. We further add:
>
> | Perturbation | Average IoU | Average F1 |
> | --- | ---: | ---: |
> | JPEG 50 | 0.292 | 0.393 |
> | JPEG 90 | 0.346 | 0.444 |
> | Resize 512x512 | 0.390 | 0.473 |
> | Gaussian Blur 3x3 | 0.379 | 0.483 |
> | Median Filter 3x3 | 0.325 | 0.421 |
>
> For operations that remove high-frequency traces (Gaussian Blur and Median Filter), performance drops but remains reasonably stable, suggesting that these operations weaken the finest texture residuals first without fully erasing the region-level discrepancy or boundary mismatch. For transmission compression, JPEG is clearly harder than resize, and heavier compression causes larger drops. Overall, these results support robustness to common transmission distortions, with meaningful but not catastrophic degradation.
>
>
> ### W2. Speed and memory footprint
>
> We benchmark the full pipeline on an NVIDIA A800 80GB PCIe GPU at 512x512, batch size 1:
>
> | Component | Mean Latency (ms) | Peak Incremental Memory |
> | --- | ---: | ---: |
> | Full pipeline | 36.2 | 128.7 MB |
> | LAD / forensic branch | 9.0 | 124.0 MB |
> | SAM refinement total | 28.4 | 84.5 MB |
> | SAM encoder only | 25.9 | 84.5 MB |
> | Adapter + decoder | 3.85 | 63.0 MB |
>
> | Post-load GPU memory | Value |
> | --- | ---: |
> | Allocated | 0.368 GB |
> | Peak allocated during end-to-end inference | 526.1 MB |
>
> These numbers show that the main overhead comes from the SAM-based refinement stage, while the LAD branch itself is lightweight. Overall, the full pipeline is fast and memory-efficient enough for practical deployment, especially for server-side forensic screening and batched analysis. We will add these measurements to make the deployment cost explicit.
>
> ### Q1. Distinguishing low-texture authentic regions
>
> We agree that low-texture authentic regions are challenging for the raw LAD response, but this does not directly imply failure of the full FLAME pipeline.
>
> 1. **How FLAME distinguishes them.** FLAME does not use LAD-Net as the final decision module: LAD-Net only provides a coarse forensic proposal, while the refinement module uses semantic consistency and boundary cues to filter many smooth but normal regions. Table 4 shows this role split: `w/o SAM` reaches 0.379 IoU / 0.458 F1, versus 0.567 / 0.662 for full FLAME, while `w/o LAD Map` drops to 0.294 / 0.380.
> 2. **Whether false positives exist.** Yes, specific false positives do exist, but mainly when both conditions hold at the same time: the authentic region is smooth or repetitive, so the LAD cue is weak or non-unique, and the edit boundary is also weak or semantically ambiguous. In these cases the coarse map becomes diffuse, and SAM 3 may not recover the correct target region. Representative examples, including false positives in smooth regions, are shown here: [failure cases](https://postimg.cc/Mn7y8s1y).
>
>
> ### Q3. LAD-Net alone versus full FLAME
>
> Table 4 provides the requested decomposition. We further summarize the decomposition below:
>
> | Variant | IoU | F1 | Interpretation |
> | --- | ---: | ---: | --- |
> | `w/o LAD Map` | 0.294 | 0.380 | semantic pipeline without explicit energy-anomaly cue |
> | `w/o Adapter` | 0.313 | 0.392 | weak forensic-semantic fusion |
> | `w/o SAM` | 0.379 | 0.458 | LAD-Net-only baseline |
> | FLAME | 0.567 | 0.662 | full coarse-to-fine model |
>
> 1. **LAD contribution.** Removing the LAD cue (`w/o LAD Map`) causes the largest overall drop, showing that LAD-Net provides the primary artifact-driven evidence.
> 2. **SAM contribution.** Comparing `w/o SAM` with full FLAME shows that SAM 3 contributes a complementary gain of +0.188 IoU / +0.204 F1, mainly through better boundary alignment and false-positive suppression.

---

> > ### Author Rebuttal · Reviewer_FDbh · 2026-04-01
> >
> > My concerns have been adequately addressed.

---

> > > ### Author Response · Authors · 2026-04-05
> > >
> > > Thank you for confirming your concerns are resolved. We would be deeply grateful if you might consider reflecting this in your final evaluation.

---

### Official Review · Reviewer_J8tf · 2026-03-12

**Soundness:** 3
**Presentation:** 3
**Significance:** 3
**Originality:** 2
**Overall Recommendation:** 4
**Confidence:** 4

**Summary:**

This paper studies pixel-level localization of AI-manipulated image edits, where traditional forensic cues such as sensor noise become unreliable. It proposes FLAME, a coarse-to-fine framework built on a Local Adjacency Discrepancy (LAD) map motivated by a Gibbs-energy view of diffusion artifacts. A lightweight LAD-Net produces image-level detection scores and coarse masks. They are then refined using a SAM 3 based adapter module. The paper also proposes the EditStream pipeline , which is a data synthesis pipeline for continuously incorporating newer editing models. Experiments report strong localization and detection performance under out-of-distribution evaluation.

**Compliance With Llm Reviewing Policy:**

Affirmed.

**Final Justification:**

The rebuttal addresses most of my concerns.

**Key Questions For Authors:**

How sensitive is LAD to naturally smooth or low-texture authentic regions, given that the paper itself notes that authentic images can contain low-energy areas that are not manipulations?

Why were recent cited detectors such as DIRE not included in the comparison set? More broadly, how would the authors position FLAME against diffusion-focused detectors beyond the selected localization baselines?

The continual adaptation story is interesting, but currently demonstrated with a single target editor and 500 fine-tuning samples. Can the authors provide evidence over a longer sequence of newly introduced editors to support the broader evolving-defense claim?

**Limitations:**

Yes

**Strengths And Weaknesses:**

Strengths:

- The method is conceptually coherent. The paper connects a theoretical argument about local energy differences in authentic versus generated regions to a concrete LAD operator, then integrates that signal into a practical coarse-to-fine architecture with LAD-Net, a feature adapter, and SAM 3 based refinement.



- FLAME achieves the best out-of-distribution average localization performance, and demonstrates  strong image-level detection performance, with further gains after EditStream fine-tuning.


Weaknesses:

- The experimental comparison set is not fully convincing given the paper’s framing around modern diffusion forensics. The baseline list includes several strong localization methods, but the paper does not compare against some relevant recent synthetic-image detectors it cites. Thus makes the positioning against the broader current literature less complete.


- The robustness study is still limited in scope. The paper evaluates only three perturbation types, each at a single relatively mild setting, and does not study resizing, heavier compression, contrast changes, or realistic social-media style pipelines that are important for deployment.

- The qualitative sections and appendix visualizations mostly showcase successful examples and intermediate maps, but there is little systematic discussion of where the method fails, for example on smooth authentic regions, very small manipulations, or highly realistic edits with weak boundary signals.

---

> ### Author Rebuttal · Authors · 2026-03-31
>
> We sincerely thank the reviewer for the constructive feedback and for recognizing FLAME's conceptual coherence, strong OOD performance, and the promise of EditStream. We add new experiments and clarifications to address the concerns below.
>
> ### W1 & Q2. Comparison to synthetic-image detectors
>
> We appreciate this question and would like to clarify the comparison protocol. FLAME targets AI-edited images, where authentic and forged regions coexist, rather than fully synthetic images. Accordingly, Table 1 compares pixel-level localizers for partial editing, not image-level synthetic-image detectors.
>
> For broader positioning, we additionally evaluate representative synthetic-image detectors on edited-image detection:
>
> | Methods   | MagicBrush ACC/AP | SIDA ACC/AP | CoCoGLIDE ACC/AP | AutoSplice ACC/AP | NanoBanana ACC/AP | Qwen ACC/AP | Flux ACC/AP | Average ACC/AP |
> | --------- | ----------------- | ----------- | ---------------- | ----------------- | ----------------- | ----------- | ----------- | -------------- |
> | DIRE (ICCV'23)      | 0.689/0.721       | 0.479/0.486 | 0.563/0.602      | 0.535/0.580       | 0.574/0.591       | 0.557/0.583 | 0.512/0.526 | 0.548/0.576    |
> | NPR (CVPR'23)       | 0.763/0.840       | 0.325/0.358 | 0.586/0.611      | 0.543/0.771       | 0.521/0.584       | 0.602/0.675 | 0.577/0.618 | 0.566/0.652    |
> | FerretNet (NeurIPS'25) | 0.891/0.952    | 0.503/0.486 | 0.623/0.684      | 0.504/0.625       | 0.632/0.681       | 0.711/0.747 | 0.662/0.693 | 0.626/0.686    |
>
> The distinction is clear: these detectors provide global discrimination, but are much less stable on edited images because large authentic regions dilute the global score. FLAME is designed for this setting and remains substantially stronger even on image-level detection (Table 2: 0.739 ACC / 0.777 AP on average). We will make this task boundary explicit in the revision.
>
> ### W2. Robustness
>
> We appreciate this concern. Beyond the perturbations already reported in Table 3, we additionally evaluate stronger JPEG compression and resizing:
>
> | Perturbation      | Average IoU | Average F1 |
> | - | - | - |
> | JPEG-50           |       0.292 |      0.393 |
> | JPEG-90           |       0.346 |      0.444 |
> | Resize to 512x512 |       0.390 |      0.473 |
>
> As expected, heavier JPEG compression causes larger degradation, while resizing is relatively milder. Overall, the performance drop is meaningful but not catastrophic.
>
> ### W3. Failure cases
>
> We appreciate this suggestion. The main failure cases arise only when the following two conditions hold simultaneously:
>
> 1. the LAD cue is weak or non-unique, such as in smooth authentic regions or scenes with repetitive local structures; and
> 2. the semantic refinement stage lacks a clear boundary prior, such as in very small edits or scenes with many similar candidate objects.
>
> In such cases, the coarse anomaly map can become diffuse, and SAM 3 may not recover the correct target region. We will add representative failure-case visualizations and discuss these limitations explicitly. Representative anonymous examples are shown here: [failure cases](https://postimg.cc/Mn7y8s1y).
>
> ### Q1. Sensitivity to naturally smooth or low-texture authentic regions
>
> We agree that naturally smooth authentic regions are challenging for the raw LAD cue, but this does not directly translate into failures of the full FLAME pipeline.
>
> 1. **How FLAME distinguishes them.** FLAME is designed as a coarse-to-fine system rather than a thresholded anomaly detector: LAD-Net proposes suspicious regions, while the SAM-based refinement stage uses semantic boundaries to reject many smooth but normal areas. Table 4 supports this role split: `w/o SAM` reaches 0.379 IoU / 0.458 F1, whereas full FLAME reaches 0.567 / 0.662; in contrast, removing the LAD cue (`w/o LAD Map`) drops to 0.294 / 0.380.
> 2. **Whether false positives exist.** Yes, specific false positives do exist, mainly when the LAD cue is weak or non-unique and the semantic boundary is also weak. These are exactly the joint failure cases discussed in W3.
>
> ### Q3. Longer sequence of newly introduced editors
>
> We appreciate this question. To better support the evolving-defense setting, we add a second adaptation step: from the Qwen-adapted FLAME-F to FLAME-FF, further adapted on Flux Kontext dataset with 500 samples.
>
> | Model | MagicBrush IoU/F1 | SIDA IoU/F1 | CoCoGLIDE IoU/F1 | AutoSplice IoU/F1 | NanoBanana IoU/F1 | Qwen IoU/F1 | Flux IoU/F1 | Average IoU/F1 |
> |-|-|-|-|-|-|-|-|-|
> | FLAME-F | 0.507 / 0.632 | 0.569 / 0.650 | 0.481 / 0.602 | 0.498 / 0.618 | 0.391 / 0.454 | 0.482 / 0.603 | 0.446 / 0.548 | 0.460 / 0.565 |
> | FLAME-FF | 0.512 / 0.637 | 0.551 / 0.640 | 0.473 / 0.598 | 0.503 / 0.629 | 0.425 / 0.473 | 0.475 / 0.599 | 0.482 / 0.591 | 0.472 / 0.578 |
>
> These results provide initial evidence for iterative adaptation over a longer editor sequence: FLAME can absorb a second editor while largely preserving earlier performance and improving on newer editors.

---

> > ### Author Rebuttal · Reviewer_J8tf · 2026-04-02
> >
> > The rebuttal addresses most of my concerns. I keep my current recommendation and score.

---

### Decision · Program_Chairs · 2026-04-30

**Decision:**

Accept (regular)

**Comment:**

The paper proposes an approach for forgery localization based on the observation that diffusion-based generation suppresses local high-frequency variance. Reviewers agree that the theoretic observation as well as the proposed method based on this observation are well motivated. Initially remaining concerns, mostly regarding the evaluation, have been fully addressed during the rebuttal phase.